# Shaping Reward with Dynamic Information for Inverse Reinforcement Learning in Stochastic Environments

## Abstract

In this paper, we aim to tackle the limitation of the Adversarial Inverse Reinforcement Learning methods in stochastic environments where theoretical results cannot hold and performance is degraded. To address this issue, we propose a novel off-policy method, based on maximum causal entropy framework, which infuses the dynamics information into the reward shaping with the theoretical guarantee for the induced optimal policy in the stochastic environments. Incorporating our novel model-based rewards, we present a novel Model-Enhanced AIRL framework, which integrates transition model estimation directly into reward shaping. Furthermore, we provide a comprehensive theoretical analysis of the reward error bound and performance difference bound for our method. The experimental results in MuJoCo benchmarks show that our method can achieve superior performance in stochastic environments and competitive performance in deterministic environments, with significant improvement in sample efficiency, compared to existing baselines.

## 1 Introduction

Reinforcement learning (RL) has achieved considerable success across various domains, including board game (Schrittwieser et al., 2020), MOBA game (Berner et al., 2019), time-delayed system (Wu et al.; 2024), and cyber-physical systems (Wang et al., 2023a;b;c; Zhan et al., 2024). Despite these advances, RL highly depends on the quality of reward function design which demands expertise, intensive labour, and a great amount of time (Russell, 1998). To address this, imitation learning (IL) methods, such as Behavior Cloning (BC) (Torabi et al., 2018a) and Inverse Reinforcement Learning (IRL) (Arora & Doshi, 2021), leverage human or expert demonstrations to bypass the need for explicit reward functions. These methods aim to learn from the demonstrations to eventually match the distribution of expert behavior, and have shown great promise in applications like autonomous driving (Codevilla et al., 2018; Sun et al., 2018), legged locomotion (Peng et al., 2020; Ratliff et al., 2009), and planning tasks (Choudhury et al., 2018; Yin et al., 2022).

The notable approaches within IRL are Adversarial Imitation Learning (AIL) methods that build upon maximum entropy framework (Ziebart et al., 2008). These adversarial methods frame imitation learning as a maximum likelihood estimation problem on trajectory distributions, converting the distribution into a Boltzmann distribution parameterized by rewards under **deterministic** environment settings (Wu et al., 2024). This closely mirrors the distribution approximation found in generative models (Finn et al., 2016a; Swamy et al., 2021). Thus, model-free AIL approaches often follow generative model structures, such as GANs (Ho & Ermon, 2016; Fu et al., 2017) or diffusion models (Reuss et al., 2023), and require extensive sampling for distribution matching gradient updates in on-policy fashion (Orsini et al., 2021). Model-based IL frameworks have also emerged, where model-based framework is designed to provide estimation for gradient and planning, leading to innovative combinations such as gradient-based IRL with model predictive control (MPC) (Das et al., 2021) and end-to-end differentiable IRL frameworks for complex robotics tasks (Baram et al., 2016; 2017; Sun et al., 2021; Rafailov et al., 2021). However, these approaches primarily address deterministic settings and struggle when applied to stochastic environments.

The only learning "deterministic" reward techniques among the existing AIL methods, rooted in their maximum entropy nature, face significant performance degeneration in stochastic environments, leading to **risk-seeking behavior** and increased data requirements (Ziebart et al., 2010). For example, an agent trained under the deterministic Markov Decision Process (MDP) might aim to imitate expert behavior by seeking high rewards, yet fail to account for the low probability of some transitions in stochastic MDP settings. This happens because, in stochastic environments, the assumption of maximum entropy that trajectory distributions are aligned with a Boltzmann distribution solely parameterized by **deterministic** rewards no longer holds. Instead, the dynamics information must also be incorporated into the formulation. There are two possible solutions. One is massive sampling to cover all possible outcomes, which is computationally expensive in large state action spaces (Devlin & Kudenko, 2011; Gupta et al., 2022). The other is changing from maximum entropy framework to maximum causal entropy framework, estimating the dynamics information, and integrating it into the reward design, making the reward "stochastic". Traditional reward design is usually based on state only $R(s_t)$ (Torabi et al., 2018b), state-action pair $R(s_t, a_t)$ (Blondé & Kalousis, 2019), or transition tuple $R(s_t, a_t, s_{t+1})$ (Fu et al., 2017), where the information inputted can be thought as a **deterministic** sample piece under the **stochastic** setting. The challenge in **stochastic** environments calls for a different perspective of rewards – stochastic rewards absorbing the transition information.

Inspired by this idea, we propose a novel maximum causal entropy based off-policy model-based adversarial IRL framework with a specifically tailored model-enhanced reward shaping approach to elevate performance in stochastic environments while remaining competitive in deterministic settings. In contrast to existing methods, our approach leverages the predictive power of the estimated transition model to shape rewards, represented as $\hat{R}(s_t, a_t, \hat{\mathcal{T}})$. This also enables us to generate synthetic trajectories to help guide policy optimization and reduce dependency on costly real-world interactions. As part of our analysis, we provide a theoretical guarantee on the optimal behavior for policies induced by our reward shaping and derive a bound on the performance gap with respect to the transition model errors. Empirically, we demonstrate that this integration significantly enhances sample complexity and policy performance in both settings, providing a comprehensive solution to the limitations of existing AIL methods in uncertain environments.

**Contributions of this work** include:

- A novel reward shaping method with model estimation under the stochastic MDP setting, which provides the optimal policy invariance guarantee.

- A novel model-based off-policy adversarial IRL framework rooted in maximum causal entropy theory that seamlessly incorporates transition model training, adversarial reward learning with model estimation and forward model-based RL process, enhancing performance in stochastic environments, and sample efficiency.

- Theoretical analysis on reward learning with model estimation under the adversarial framework and performance difference under transition model learning errors.

- Empirical validation that demonstrates our approach's performance improvements in stochastic environments as well as significant improvement in sample efficiency and comparable performance in deterministic environments.

In Sec. 2, we introduce related works in AIL and reward shaping. In Sec. 3, we provide the necessary preliminaries for MDP and IRL. In Sec. 4, we present our model-enhanced reward shaping method and corresponding theoretical guarantee. In Sec. 5, we present our Model-Enhanced AIRL framework design together with derivation from maximum causal entropy objective, theoretical analysis on reward error bound, and performance difference bound. In Sec. 6, we show the experimental results in `Mujoco` for various benchmarks. Sec. 7 concludes the paper.

## 2 RELATED WORKS

**Adversarial Imitation Learning.** Margin optimization based IRL methods (Ng et al., 2000; Abbeel & Ng, 2004; Ratliff et al., 2006) aim to learn reward functions that explain expert behavior better than other policies by a margin. Bayesian approaches were introduced with different prior assumptions on reward distributions, such as Boltzmann distributions (Ramachandran & Amir, 2007; Choi & Kim, 2011; Chan & van der Schaar, 2021) or Gaussian Processes (Levine et al., 2011). Other

statistical learning methods include multi-class classification (Klein et al., 2012; Brown et al., 2019) and regression trees (Levine et al., 2010). The entropy optimization approach has seen significant development. To avoid biases from maximum margin methods, the maximum entropy principle (Shore & Johnson, 1980) is used to infer distributions over trajectories parameterized by reward weights. Ziebart et al. (2008; 2010) proposed a Lagrangian dual framework to cast the reward learning into a maximum likelihood problem with linear-weighted feature-based reward representation. Wulfmeier et al. (2015) extended the framework to nonlinear reward representations, and Finn et al. (2016b) combines importance sampling techniques to enable model-free estimation. Inspired by GANs, adversarial methods were introduced for policy and reward learning in IRL (Ho & Ermon, 2016; Fu et al., 2017; Torabi et al., 2018b). However, **these methods typically work with Maximum Entropy (ME) formulation yet suffer from sample inefficiency and stochasticity**. Although there have been efforts to combine adversarial methods with off-policy RL agents to improve sample efficiency (Kostrikov et al., 2018; Blondé & Kalousis, 2019; Blondé et al., 2022), few extend it to the model-based setting which might further the improvement, and none of these approaches addresses the rewards learning in stochastic MDP settings.

**Rewards Shaping.** Reward shaping (Dorigo & Colombetti, 1994; Randløv & Alstrøm, 1998) is a technique that enhances the original reward signal by adding additional domain information, making it easier for the agent to learn optimal behavior. This can be defines as $\hat{R} = R + F$, where $F$ is the shaping function and $\hat{R}$ is the shaped reward function. Potential-based reward shaping (PBRS) (Ng et al., 2000) builds the potential function on states, $F(s, a, s') = \phi(s') - \phi(s)$, while ensuring the policy invariance property, which refers to inducing the same optimal behavior under different rewards $R$ and $\hat{R}$. Nonetheless, there also exist other variants on the inputs of the potential functions such as state-action (Wiewiora et al., 2003), state-time (Devlin & Kudenko, 2012), and value function (Harutyunyan et al., 2015) as potential function input. There are also some latest attempts of reward shaping without utilization of domain knowledge potential function to solve exploration under sparse rewards (Hu et al., 2020; Devidze et al., 2022; Gupta et al., 2022; Skalse et al., 2023).

**MBIRL.** Model-Based RL (MBRL) has emerged as a promising direction for improving sample efficiency and generalization (Janner et al., 2019; Yu et al., 2020). MBRL combines various learned dynamics neural network structures with planning (Hansen et al., 2022; Sikchi et al., 2022).This framework has been successfully extended to vision-based control tasks (Hafner et al., 2019; Zhan et al., 2024). Integrating IRL with MBRL has also shown success. For example, Das et al. (2021) and Herman et al. (2016) presented a gradient-based IRL approach using different policy optimization methods with dynamic models for linear-weighted features reward learning. In Das et al. (2021), the dynamic model is used to pass forward/backward the gradient in order to update the IRL and policy optimization modules. Similarly, end-to-end differentiable adversarial IRL frameworks to various state spaces have also been explored (Baram et al., 2016; 2017; Sun et al., 2021; Rafailov et al., 2021), where dynamic model serves a similar role. Despite these advancements, existing methods rarely address the specific challenges posed by stochastic environments, which limit reward learning performance. To our knowledge, this is the first study that provide a theoretical analysis on the performance difference with learned dynamic model for the adversarial IRL problem under stochastic MDP.

## 3 PRELIMINARIES

**MDP.** RL is usually formulated as a Markov Decision Process (MDP) $\mathcal{M}$ (Puterman, 2014) denoted as a tuple $\langle \mathcal{S}, \mathcal{A}, \mathcal{T}, \gamma, R, \rho_0 \rangle$. $\rho_0$ is the initial distribution of the state. $s \in \mathcal{S}, a \in \mathcal{A}$ stands for the state and action space respectively. $\mathcal{T}$ stands for the transition dynamic such that $\mathcal{T} : \mathcal{S} \times \mathcal{A} \times \mathcal{S} \to [0, 1]$. $\gamma \in (0, 1)$ is the discounted factor, $R$ stands for reward function such that $R : \mathcal{S} \times \mathcal{A} \to \mathbb{R}$ and $\|R\|_\infty \le R_{\max}$. The discounted visitation distribution of trajectory $\tau$ with policy $\pi$ is given by:

$$p(\tau) = \rho_0 \prod_{t=0}^{T-1} \gamma^t \mathcal{T}(s_{t+1}|s_t, a_t)\pi(a_t|s_t). \tag{1}$$

The objective function of RL is $\max \mathbb{E}_{\tau \sim p(\tau)} \left[ \sum_{t=0}^{T} \gamma^t R(\tau) - H(\pi) \right]$, where $H$ is the log likelihood of the policy. We introduce Soft Value Iteration for bellmen update (Haarnoja et al., 2018), where $Q^{soft}$ and $V^{soft}$ denotes the soft Q function and Value function respectively:

$$V^{soft}(s_t) = log \sum_{a_t \in \mathcal{A}} \exp Q^{soft}(s_t, a_t) da_t, \tag{2}$$

$$Q^{soft}(s_t, a_t) = R(s_t, a_t) + \gamma \mathbb{E}_{\mathcal{T}} \left[ V^{soft}(s_{t+1}) | s_t, a_t \right], \tag{3}$$

$$\pi(a_t | s_t) = \exp \left( Q^{soft}(s_t, a_t) - V^{soft}(s_t) \right), \tag{4}$$

where the soft Advantage function is defined as $A^{soft}(s_t, a_t) = Q^{soft}(s_t, a_t) - V^{soft}(s_t)$.

**Inverse RL.** In IRL setting, we usually consider the MDP without reward as $\mathcal{M}'$ where $R$ is also unknown. We denote the data buffer $\mathcal{D}_{exp}$ which collects trajectories from an expert policy $\pi^E$. We consider a reward function $R_\theta : \mathcal{S} \times \mathcal{A} \to \mathbb{R}$, where $\theta$ is the reward parameter. An IRL problem can be defined as a pair $\mathcal{B} = (\mathcal{M}', \pi^E)$. A reward function $R_\theta$ is feasible for $\mathcal{B}$ if $\pi^E$ is an optimal policy for the MDP $\mathcal{M}' \cup R_\theta$, and we denote the set of feasible rewards as $\mathcal{R}_\mathcal{B}$. Using maximize likelihood estimation framework, we can formulate the IRL as the following maximum causal entropy problem:

$$\arg \max_\theta \mathbb{E}_{\tau \sim \mathcal{D}_{exp}} \log p_\theta(\tau), \tag{5}$$

where $Q_{R_\theta}^{soft}$ and $V_{R_\theta}^{soft}$ are based on $R_\theta$ and $p_\theta(\tau) \propto \rho_0 \prod_{t=0}^{T-1} \mathcal{T}(s_{t+1}|s_t, a_t) \exp(Q_{R_\theta}^{soft}(s_t, a_t) - V_{R_\theta}^{soft}(s_t))$ (Ziebart et al., 2010). Under **deterministic** MDP, the above problem can be simplified as ME problem, where $p_\theta(\tau) \propto \frac{1}{Z_\theta} \exp \sum_{t=0}^{T-1} R_\theta(s_t, a_t)$ and $Z_\theta$ is the temperature factor of the Boltzmann Distribution (Ziebart et al., 2008).

## 4 MODEL ESTIMATION IN REWARD SHAPING

Table 1: We summarize the different reward formulations and their dynamic properties in this table. Components refer to the input pair that the reward functions take. Reward Shaping indicates whether there is the extra physical potential information involved where X means no reward shaping used. Dynamics information shows whether transitions are involved in the reward function.

| Methods | Components | Reward Shaping | Dynamics Information |
|---|---|---|---|
| AIRL (Fu et al., 2017) | $s_t, a_t, s_{t+1}$ | $R(s_t, a_t) + \gamma \phi(s_{t+1}) - \phi(s_t)$ | single sample |
| AIRL(State Only) | $s_t$ | $R(s_t) + \text{constant}$ | X |
| DAC (Kostrikov et al., 2018) | $s_t, a_t$ | X | X |
| SAM (Blondé & Kalousis, 2019) | $s_t, a_t$ | X | X |
| SQIL (Reddy et al., 2019) | $s_t, a_t$ | binary | X |
| GAIfO (Torabi et al., 2018b) | $s_t, s_{t+1}$ | X | single sample |
| Ours | $s_t, a_t, \mathcal{T}$ | $R(s_t, a_t) + \gamma \mathbb{E}_\mathcal{T}[\phi(s_{t+1})|s_t, a_t] - \phi(s_t)$ | transition model |

In this section, we illustrate the advantages of involving transition dynamics into the reward shaping, especially in stochastic MDP settings. Most of literature work has various formulations and definitions (Table 1), but few considers transition dynamic information in the reward shaping. Defining rewards solely based on states, $R^s(s_t)$, offers limited utility in environments where actions are critical. Even though the state-action pair-based rewards $R^{sa}(s_t, a_t)$ can capture the missing information of the taken action, it fails to consider any future information, the successive state $s_{t+1}$. Transition tuple-based rewards $R^{tuple}(s_t, a_t, s_{t+1})$ incorporate the dynamics information in a sampling-based way, which requires abundant data to learn the underlying relationship of two consecutive states, potentially raising the sample efficiency issue in the stochastic environment with the huge state space. To address this issue, we propose dynamics-based rewards shaping $\hat{R}(s_t, a_t, \mathcal{T})$, which explicitly infuse the dynamics information $\mathcal{T}$ on the potential function, thus significantly improving the sample-efficiency. Specifically, our rewards shaping is defined as

$$\hat{R}(s_t, a_t, \mathcal{T}) = R(s_t, a_t) + \gamma \mathbb{E}_\mathcal{T} \left[ \phi(s_{t+1}) | s_t, a_t \right] - \phi(s_t), \tag{6}$$

where $\phi$ is a state-only potential function, $\mathcal{T}$ is the dynamics. Another insight of the above reward shaping is to resemble the advantage function with the soft value function as the potential function,

which we will elaborate in Sec. 5.1. With the given reward shaping $\hat{R}$, it is crucial to show that it induces the same optimal behaviour as the ground-true reward $R$. We formally define this policy invariance property as follows.

**Definition 4.1.** *(Memarian et al., 2021) Let $R$ and $\hat{R}$ be two reward functions. We say they induce the same soft optimal policy under transition dynamics $\mathcal{T}$ if, for all states $s \in \mathcal{S}$ and actions $a \in \mathcal{A}$:*

$$A_R^{soft}(s_t, a_t) = A_{\hat{R}}^{soft}(s_t, a_t). \tag{7}$$

With the above definition, we can transfer the proof of policy invariant property of our designed reward shaping (Eq. (6)) to showing the equivalence of soft advantage functions, which is proved in the following theorem. The detailed proof can be found in A.1.

**Theorem 4.2** (Policy Invariance). *Let $R$ and $\hat{R}$ be two reward functions. $R$ and $\hat{R}$ induce the same soft optimal policy under all transition dynamics $\mathcal{T}$ if $\hat{R}(s_t, a_t, \mathcal{T}) = R(s_t, a_t) + \gamma \mathbb{E}_{\mathcal{T}}[\phi(s_{t+1})|s_t, a_t] - \phi(s_t)$ for some potential-shaping function $\phi : \mathcal{S} \to \mathbb{R}$.*

Thm. 4.2 implies that the optimal policy induced from our model-enhanced rewards shaping $\hat{R}$ (Eq. (6)) is equivalent to the optimal policy trained by the ground-truth reward function $R$ under the soft Value Iteration fashion.

## 5 MODEL ENHANCED ADVERSARIAL IRL

In this section, we first elaborate on the adversarial formulation of our reward shaping (Eq. (8)) and present the theoretical insight (Proposition 5.1) of the equivalence between cross-entropy training loss of adversarial reward shaping formulation and maximum log-likelihood loss of original maximum causal entropy IRL problem. Then, in the Sec. 5.2, we showcase our practical algorithm framework with trajectory generation and transition model learning in the loop, as shown in Fig. 1. Furthermore, we theoretically investigate the reward function bound (Thm. 5.3) and performance difference bound (Thm. 5.4) under the transition model learning error.

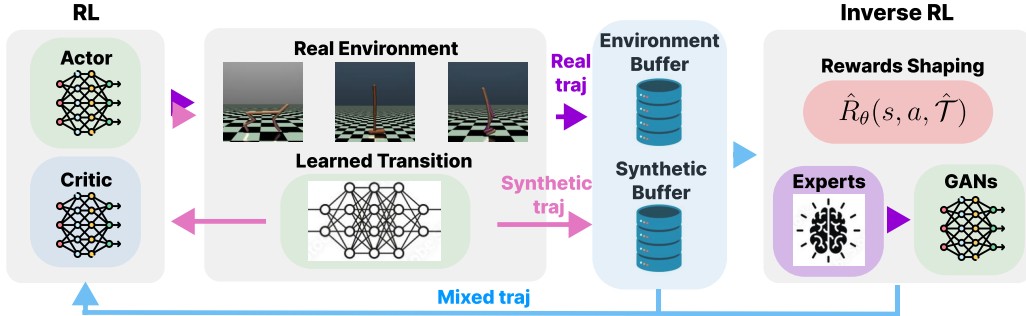

Figure 1: Framework overview of Model-Enhanced Adversarial IRL. Different color arrows stand for different sample flows. Purple stands for real environmental interaction samples, pink stands for synthetic samples generated from learned transition model, and blue stands for mixed of both.

### 5.1 ADVERSARIAL FORMULATION OF REWARD SHAPING

In this section, we connect the reward shaping in adversarial training framework with rewards learning objective under the MCE framework. Inspired by GANs (Goodfellow et al., 2014), the idea behind adversarial framework is to train a binary discriminator $D(s_t, a_t, s_{t+1})$ or $D(s_t, a_t)$ to distinguish state-action-transition samples from an expert and those generated by imitator policy following the original ME setting. However, as mentioned above, we only take in state-action pair and transition function to define our reward function which also extends to our discriminator as follows:

$$D_\theta(s_t, a_t, \mathcal{T}) = \frac{\exp\{f_\theta(s_t, a_t, \mathcal{T})\}}{\exp\{f_\theta(s_t, a_t, \mathcal{T})\} + \pi(a_t|s_t)}, \tag{8}$$

---

**Algorithm 1** Model Enhanced Adversarial IRL

---

1: Obtain expert buffer $\mathcal{D}_{exp}$.
2: Initialize policy $\pi$, discriminator $D_\theta$, buffers $\mathcal{D}_{env}, \mathcal{D}_{gen}$, and transition model $\hat{\mathcal{T}}$.
3: **for** step $t$ in $\{1, \ldots, N\}$ **do**
4:     Interact with real environments and add state-action pair to $\mathcal{D}_{env}$.
5:     **if** $t <$ STARTING_STEP **then**
6:         Pretrain transition model $\hat{\mathcal{T}}$.
7:     **else**
8:         Sample state-action batch from $\mathcal{D}_{exp}, \mathcal{D}_{env}$ respectively.
9:         Train $D_\theta$ via cross entropy loss Eq. (9) to classify expert data from samples.
10:        Update dynamic model $\hat{\mathcal{T}}$ with MLE loss and generate $H$-steps trajectories to $\mathcal{D}_{gen}$.
11:        Sample state-action batches from $\mathcal{D}_{env}$, and $\mathcal{D}_{gen}$ with varying ratio.
12:        Update $\pi$ with respect to $\hat{R}_\theta$ using Soft Actor Critic policy optimization.
13:     **end if**
14: **end for**

---

where $f_\theta(s_t, a_t, \mathcal{T}) = R_\theta(s_t, a_t) + \gamma \mathbb{E}_\mathcal{T}\left[\phi_\theta(s_{t+1})|s_t, a_t\right] - \phi_\theta(s_t)$ resembles the reward shaping defined above. The loss function for the training discriminator is defined below.

$$\mathcal{L}_{disc} = -\mathbb{E}_{\mathcal{D}_{exp}}\left[\log D_\theta(s, a, \mathcal{T})\right] - \mathbb{E}_\pi\left[\log(1 - D_\theta(s, a, \mathcal{T}))\right]. \tag{9}$$

We bridge this adversarial formulation with the original MCE IRL problem. In the following proposition, we give a sketch of proof to show the connection between the objective function of discriminator and MCE IRL. Proof details can be found in Appendix B.1.

**Proposition 5.1.** *Consider an undiscounted MDP. Suppose $f_\theta$ and $\pi$ at the current iteration are the soft-optimal advantage function and policy for reward function $R_\theta$. Minimising the cross-entropy loss of the discriminator under generator $\pi$ is equivalent to maximising the log-likelihood under Maximum Causal Entropy IRL.*

With above given proposition, we can construct a direct intuition that $f_\theta^*$ should be equal to $\hat{R}_\theta$ the reward shaping we introduced early and resemble the soft advantage function. To extract rewards to represent reward used for policy optimization, we use $\log(D_\theta(s, a, \mathcal{T})) - \log(1 - D_\theta(s, a, \mathcal{T}))$, which resembles the entropy-regularized reward shaping $f_\theta(s, a, \mathcal{T}) - \log \pi(a|s)$. Given this entropy-regularized reward, it is straightforward to see why the optimal policy can satisfy the RL objectives.

## 5.2 ALGORITHM FRAMEWORK

In this section, we present the overall framework of Model-Enhanced Adversarial IRL and illustrate how transition model training is incorporated into the learning loop. We assume the estimated transition distribution $\hat{\mathcal{T}}(\cdot|s, a)$ follows a Gaussian distribution with mean and standard deviation parameterized by the MLP, and the model is updated with standard maximum likelihood loss. The transition model is updated in each policy optimization iteration similar as model-based RL approaches (Janner et al., 2019; Hansen et al., 2022; Zhan et al., 2024). At each iteration, the updated transition model is utilized for reward learning and synthetic data generation in eval mode, which is stored in the synthetic trajectory replay buffer. Unlike AIRL and GAIL, our framework operates in an **off-policy** fashion, where samples used for both discriminator and policy update are drawn from a combination of the environmental replay buffer and the synthetic replay buffer. An overview of our framework is shown in Fig. 1, and detailed algorithmic steps and parameters are provided in Alg. 1 and Appendix F.

**Sample Efficiency:** To improve sample efficiency, we leverage the estimated transition model to generate $H$-steps synthetic trajectories data alongside real interaction data, facilitating policy optimization. Given that the estimated transition model is inaccurate at the beginning, we employ a dynamic ratio between real and synthetic data to prevent the model from being misled by unlikely synthetic transitions (Janner et al., 2019; Zhan et al., 2024). Specifically, early-stage generated trajectories are not stored persistently, unlike real interactions which are fully stored in the off-policy environmental replay buffer. To maintain training stability, we use a synthetic replay buffer with a size that gradually increases as training progresses, ensuring a balanced inclusion of synthetic data

over time. The growth rates of the data ratio and buffer size are adjusted based on the complexity of the transition model learning process and can be fine-tuned via hyper-parameters. *H* horizon choosing and buffer size update scheme can be found in Appendix F.

**Distribution Shift:** To mitigate distribution shift (Lee et al., 2020; Lin et al., 2020) during training, we employ a strategy involving the learned transition model. Typically, during interaction, the real state $s_t$ is used as input to the actor, and the resulting action $a_t$ is applied in the environment. To incorporate the transition model, we predict a synthetic state $\hat{s}_t$ from previous $s_{t-1}$ and $a_{t-1}$. This generated $\hat{s}_t$ is then fed into the actor to produce action $\hat{a}_t$. The actions $a_t$ and $\hat{a}_t$ are mixed and applied to the environment with a certain ratio, and the resulting pairs $(s_t, a_t)$ or $(s_t, \hat{a}_t)$ are stored in the environmental replay buffer. This approach helps balance the exploration of real and model-predicted dynamics, reducing the impact of distributional discrepancies.

## 5.3 PERFORMANCE ANALYSIS

In this section, we analyze the optimal performance bound in the presence of transition model learning errors. Our results show that as the transition model error approaches zero, the performance gap at the optimal point vanishes at the same time. The learned transition model $\hat{\mathcal{T}}$ persists in some errors compared with the ground-true transition dynamic. In this section, we investigate how this error will affect performance of our method. As a reminder, we define an IRL problem as $\mathfrak{B} = (\mathcal{M}', \pi^E)$, where $\mathcal{M}'$ is a MDP without $R$ and $\pi^E$ is an optimal expert policy. We denote $\mathcal{R}_\mathfrak{B}$ as the set of feasible rewards set for $\mathfrak{B}$. Since under our case $\mathcal{T}$ is approximated by $\hat{\mathcal{T}}$, we have another IRL problem defined as $\hat{\mathfrak{B}} = (\hat{\mathcal{M}}', \pi^E)$ where $\hat{\mathcal{M}}'$ has the same state and action space, discount factor, and initial distribution but an estimated transition model $\hat{\mathcal{T}}$. For notation, we use $D_{\text{TV}}$ to denote the total variation distance, $\| \cdot \|$ to represent the infinity norm (with $\infty$ omitted for simplicity), $|\mathcal{S}|$ to denote the cardinality of the state space, and $V_{\mathcal{M}' \cup R}^{\pi^*}$ to represent the value function of policy $\pi^*$ under the MDP $\mathcal{M}'$ with reward $R$, and vice versa.

**Assumption 5.2** (Transition Model Error). *Since transition model is trained through a supervised fashion, we can use a PAC generalization bound (Shalev-Shwartz & Ben-David, 2014) for sample error. Therefore, we assume that the total variation distance between $\mathcal{T}$ and $\hat{\mathcal{T}}$ is bounded by $\epsilon_\mathcal{T}$ through $[0, T]$:*

$$\max_t \mathbb{E}_{s \sim \pi_{D,t}} \left[ D_{TV}(\mathcal{T}(s'|s,a)|\hat{\mathcal{T}}(s'|s,a)) \right] \leq \epsilon_\mathcal{T}, \tag{10}$$

*which is a common assumption that adopted in literature (Janner et al., 2019; Sikchi et al., 2022).*

Next, with the assumed total visitation bound on transition models ( Assumption 5.2), we want to reflect this bound to the error in rewards learning through our model-enhanced reward shaping.

**Theorem 5.3** (Reward Function Error Bound). *Let $\mathfrak{B} = (\mathcal{M}', \pi^*)$ and $\hat{\mathfrak{B}} = (\hat{\mathcal{M}}', \pi^*)$ be two IRL problems with transition functions $\mathcal{T}$ and $\hat{\mathcal{T}}$ respectively, then for any $R^E \in \mathcal{R}_\mathfrak{B}$ there is a corresponding $\hat{R}^E \in \mathcal{R}_{\hat{\mathfrak{B}}}$ such that*

$$\|R^E - \hat{R}^E\| \leq \frac{\gamma}{1-\gamma}|\mathcal{S}|\epsilon_\mathcal{T} R_{\max}. \tag{11}$$

Proof of Thm. 5.3 can be found in Appendix C.3. With rewards bound above, we can extend the bound to the value function, which represents the performance difference brought up by estimated transition model error under RL setting.

**Theorem 5.4** (Performance Difference Bound). *The performance difference between the optimal policies ($\pi^*$ and $\hat{\pi}^*$) in corresponding MDPs ($\mathcal{M}' \cup R$ and $\hat{\mathcal{M}}' \cup \hat{R}$) can be bounded as follows:*

$$\|V_{\mathcal{M}' \cup R^E}^{\pi^*} - V_{\hat{\mathcal{M}}' \cup \hat{R}^E}^{\hat{\pi}^*}\| \leq \epsilon_\mathcal{T} \left[ \frac{\gamma}{(1-\gamma)^2} R_{\max} + \frac{1+\gamma}{(1-\gamma)^2} R_{\max}|\mathcal{S}| \right]. \tag{12}$$

The detailed proof of Thm. 5.4 is presented in Appendix C.6. The above theorem highlights the relationship between the performance gap and the transition model error, also implying that a perfectly-learned transition model ($\epsilon_\mathcal{T} \to 0$) could make the performance difference negligible.

Table 2: Best performance of expert and all algorithms in deterministic MuJoCo Environments under conditions of different numbers of expert trajectories provided (10, 100, and 1000). AIRL and GAIL are trained with `10M` environmental steps. DAC and Our are trained with `1M` environmental steps.

| Environment | Expert Trajs | Expert | GAIL | AIRL | DAC | Ours |
|---|---|---|---|---|---|---|
| InvertedPendulum-v4 | 10 | $1000.0_{\pm 0.0}$ | $1000.0_{\pm 0.0}$ | $1000.0_{\pm 0.0}$ | $1000.0_{\pm 0.0}$ | $1000.0_{\pm 0.0}$ |
| | 100 | $1000.0_{\pm 0.0}$ | $1000.0_{\pm 0.0}$ | $1000.0_{\pm 0.0}$ | $1000.0_{\pm 0.0}$ | $1000.0_{\pm 0.0}$ |
| | 1000 | $986.09_{\pm 95.97}$ | $1000.0_{\pm 0.0}$ | $1000.0_{\pm 0.0}$ | $1000.0_{\pm 0.0}$ | $1000.0_{\pm 0.0}$ |
| InvertedDoublePendulum-v4 | 10 | $131.3_{\pm 77.0}$ | $155.0_{\pm 58.1}$ | $163.0_{\pm 48.6}$ | $100.6_{\pm 11.8}$ | $193.4_{\pm 15.5}$ |
| | 100 | $108.0_{\pm 43.2}$ | $167.2_{\pm 26.6}$ | $151.2_{\pm 28.6}$ | $94.5_{\pm 9.9}$ | $198.1_{\pm 76.3}$ |
| | 1000 | $140.44_{\pm 76.62}$ | $189.5_{\pm 28.8}$ | $150.2_{\pm 18.3}$ | $105.6_{\pm 20.4}$ | $182.2_{\pm 29.6}$ |
| Hopper-v4 | 10 | $1786.0_{\pm 803.0}$ | $1266.9_{\pm 366.2}$ | $2092.2_{\pm 57.4}$ | $1000.4_{\pm 5.3}$ | $2408.4_{\pm 641.7}$ |
| | 100 | $1489.6_{\pm 659.6}$ | $2385.9_{\pm 350.0}$ | $2789.9_{\pm 30.8}$ | $993.1_{\pm 10.5}$ | $2820.9_{\pm 89.8}$ |
| | 1000 | $1516.0_{\pm 692.6}$ | $2746.5_{\pm 270.9}$ | $2744.3_{\pm 37.4}$ | $2007.1_{\pm 719.7}$ | $2858.8_{\pm 76.9}$ |
| HalfCheetah-v4 | 10 | $1567.4_{\pm 74.1}$ | $368.5_{\pm 53.7}$ | $463.9_{\pm 61.2}$ | $9.5_{\pm 457.2}$ | $888.6_{\pm 67.3}$ |
| | 100 | $1120.5_{\pm 67.5}$ | $398.1_{\pm 123.5}$ | $556.0_{\pm 12.8}$ | $615.9_{\pm 250.5}$ | $1108.3_{\pm 13.9}$ |
| | 1000 | $1113.5_{\pm 76.1}$ | $735.6_{\pm 44.0}$ | $708.7_{\pm 14.5}$ | $1046.4_{\pm 13.9}$ | $1162.8_{\pm 62.2}$ |
| Walker2d-v4 | 10 | $3109.4_{\pm 1031.5}$ | $1262.8_{\pm 396.3}$ | $1170.5_{\pm 484.0}$ | $101.4_{\pm 149.1}$ | $2509.0_{\pm 860.0}$ |
| | 100 | $3295.4_{\pm 704.0}$ | $956.4_{\pm 313.2}$ | $1740.7_{\pm 609.8}$ | $416.1_{\pm 243.2}$ | $3311.0_{\pm 157.2}$ |
| | 1000 | $3268.9_{\pm 746.1}$ | $1430.6_{\pm 489.8}$ | $3051.3_{\pm 210.5}$ | $3531.3_{\pm 105.3}$ | $3497.8_{\pm 51.7}$ |

## 6 EXPERIMENTS

In this section, we evaluate the performance and sample efficiency of our Model-Enhanced Adversarial IRL framework. We aim to demonstrate the superiority of our method in stochastic environments, achieving better performance and sample efficiency compared to existing approaches. Additionally, in deterministic settings, our method maintains competitive performance with baselines. All experiments are conducted on the MuJoCo benchmarks (Todorov et al., 2012). To simulate stochastic dynamics in MuJoCo, we introduce the agent-unknown Gaussian noise with a mean of 0 and a standard deviation of 0.5 to the environmental interaction steps. All the **expert trajectories** are collected by an expert agent trained with standard SAC (Haarnoja et al., 2018) under deterministic or stochastic MuJoCo environments. Our experiments are designed to highlight the key advantages of our framework:

- **Performance in Stochastic Environments:** In stochastic settings, our method significantly outperforms other approaches, consistently surpassing expert-level performance more rapidly. This enhanced ability to learn under uncertainty is attributed to our framework's effectiveness in leveraging model-based predictiton capability.

- **Sample Efficiency** For stochastic settings, our method can reach expert performance with fewer training steps than all the other baselines with various conditions on expert demonstrations provided. Besides, we showcase that our method can extract reward signal from few expert demonstrations under the stochastic setting, which majority of the baseline failed.

- **Performance in Deterministic Environments:** We demonstrate that our method is competitive with existing AIL methods' performances in deterministic settings.

We primarily compare our approach with other Adversarial Imitation Learning (AIL) methods, including the on-policy algorithms GAIL (Ho & Ermon, 2016) and AIRL (Fu et al., 2017), and the off-policy method Discriminator Actor-Critic (DAC) (Kostrikov et al., 2018). For policy optimization, we use Proximal Policy Optimization (PPO) (Schulman et al., 2017) for both GAIL and AIRL, and Soft Actor-Critic (SAC) (Haarnoja et al., 2018) for DAC. All implementations of PPO and SAC are referenced from the `Clean RL` library (Huang et al., 2022). Each algorithm is trained with 100k environmental steps and evaluated each 1k steps across 5 different seeds for tasks including `InvertedPendulum-v4` and `InvertedDoublePendulum-v4`. For `Hopper-v4`, `HalfCheetah-v4, and `Walker2d-v4`, AIRL and GAIL are trained with 10M steps and evaluated each 100k steps across 5 different seeds, but DAC and our algorithm are trained with 1M environmental steps and evaluated each 10k steps across 5 different seeds. We conduct aforementioned series of experiments under **various numbers of expert trajectories ranging from 5 to 1000**. All the experiments are run on the Desktop equiped with RTX 4090 and Core-i9 13900K. The learning curves of all methods are provided in Appendix D.

**Performance in Stochastic MuJoCo.** In Table 2, we present the performance of our method and baseline methods in stochastic MuJoCo environments with varying numbers of expert trajectories.

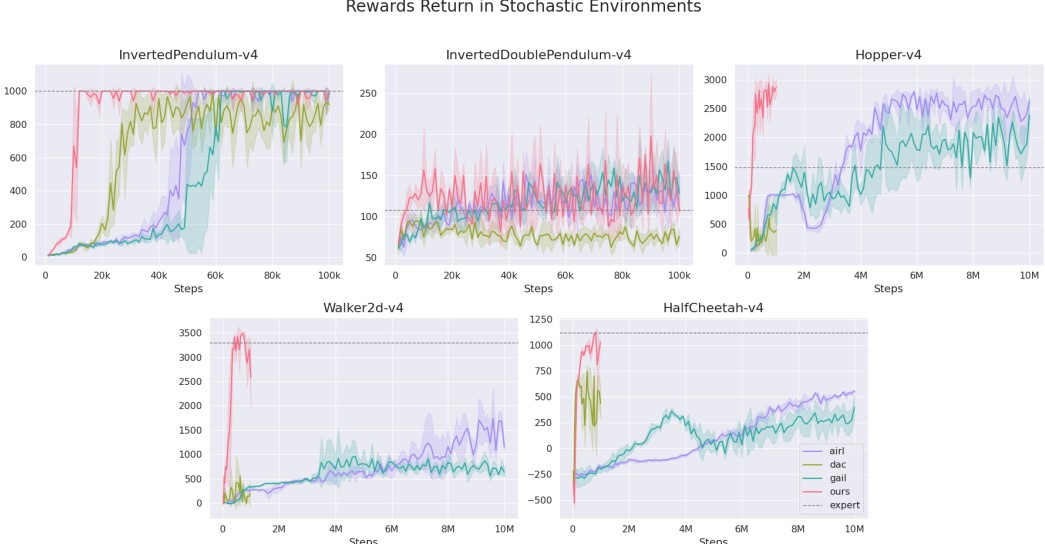

Figure 2: Training curves of all 4 methods in 5 different **stochastic** environments with 100 expert trajectories. For better comparison in sample efficiency, graph is presented under `10M` landscape.

**Our method consistently achieves the best performance across the majority of these environments, outperforming all baselines under different levels of expert trajectory availability**. In simpler environments, such as `InvertedPendulum-v4`, the introduction of stochasticity and variations in expert trajectory have minimal impact on the final performance for both our method and the baselines. However, for more complex environments, the effect of stochasticity becomes more pronounced. Specifically, in `InvertedDoublePendulum-v4`, stochasticity notably degrades performance. Our method, however, maintains a competitive edge over all baselines, achieving better results with limited expert trajectories (10 and 100) and reaching similar performance to the baselines when more expert trajectories are available. In `Hopper-v4`, our method substantially outperforms all baselines, especially when fewer expert trajectories are provided. As the number of expert demonstrations increases, the performance gap between our method and the baselines narrows due to the growing reference sample size. Nonetheless, our method maintains an edge in sample efficiency, which we will discuss further in the next paragraph. Similar performance trends are observed in environments such as `HalfCheetah-v4` and `Walker2d-v4`. These results indicate that our approach can effectively recover the reward function more closely from demonstrations in stochastic environments, resulting in significant performance improvement. Additionally, in the stochastic settings, the performance of DAC decreases significantly, due to DAC's ineffective reward formulation on state-action pairs discussed in Sec. 4, which also result in training instability shown in Appendix D.

**Sample Efficiency.** In Appendix D, we display the sample efficiency across various environments and with different numbers of expert trajectories. Since AIRL and GAIL use distinct environmental training steps from DAC and our method, we provide a clearer comparison in Fig. 2. Based on results, **our method shows significant superiority in sample efficiency across all of the benchmarks under the stochastic settings. Additionally, our method also demonstrate significant advantage when limited expert trajectories are available.** Specifically, for `InvertedPendulum-v4`, as shown in Fig. 3, all methods can achieve expert-level performance except DAC, which exhibits instability with limited demonstrations. Our method, however, consistently reaches expert-level performance in the fewest training steps, regardless of the number of expert trajectories. In `InvertedDoublePendulum-v4` as shown in Fig. 4, introducing stochasticity into the dynamics makes it challenging for all algorithms to achieve reasonable performance from noisy expert demonstrations. Notably, DAC completely fails to reach expert-level performance, whereas our method attains it with the fewest training steps across all levels of expert trajectory availability. For `Hopper-v4` in Fig. 5, our method is the only approach capable of reaching expert-level performance

Table 3: Best performance of expert and all algorithms in deterministic MuJoCo Environments with 1000 expert trajectories provided. DAC and our methods are trained for `1M` environmental steps. GAIL and AIRL are trained for `10M` environmental steps.

| Environment | Expert | GAIL | AIRL | DAC | Ours |
|---|---|---|---|---|---|
| InvertedPendulum-v4 | $1000.0_{\pm 0.0}$ | $1000.0_{\pm 0.0}$ | $1000.0_{\pm 0.0}$ | $1000.0_{\pm 0.0}$ | $1000.0_{\pm 0.0}$ |
| InvertedDoublePendulum-v4 | $9356.7_{\pm 0.2}$ | $9324.4_{\pm 0.4}$ | $355.3_{\pm 76.3}$ | $9359.8_{\pm 0.1}$ | $9359.8_{\pm 0.1}$ |
| Walker2d-v4 | $4520.7_{\pm 648.44}$ | $3387.0_{\pm 617.8}$ | $3623.6_{\pm 189.6}$ | $4655.3_{\pm 126.4}$ | $4396.7_{\pm 147.4}$ |
| Hopper-v3 | $3262.8_{\pm 314.4}$ | $3420.8_{\pm 77.9}$ | $3385.8_{\pm 50.5}$ | $3481.6_{\pm 94.6}$ | $3506.6_{\pm 23.5}$ |
| HalfCheetah-v3 | $13498.6_{\pm 710.9}$ | $3502.6_{\pm 202.3}$ | $3237.8_{\pm 85.5}$ | $10102.2_{\pm 297.6}$ | $6509.8_{\pm 177.7}$ |

consistently even with limited number of expert trajectories. GAIL and AIRL borh fail to reach the expert within `1M` environmental training steps. DAC was able to reach expert performance only when expert trajectories are sufficient, though it still suffers from sample inefficiency and training instability. Similar trends can also be observed in `HalfCheetah-v4` (Fig. 6) and `Walker2d-v4` (Fig. 7). We also observe a universal trend across all stochastic environments: as the number of expert trajectories increases, both the sample efficiency and performance of all methods improve accordingly, which aligns with intuitive expectations.

**Performance in Deterministic MuJoCo.** The performance of deterministic MuJoCo environments can be found in Table 3. For the tasks with deterministic dynamics, our method can achieve the performance aligning with all of baselines and the expert in `InvertedDoublePendulum-v4, InvertedDoublePendulum-v4, Hopper-v4,` and `Walker2d-v4`. For `HalfCheetah-v4`, our method has exceeding performance comparing with AIRL and GAIL, but fail to reach the similar level as DAC and expert. Since as the dynamic becomes complicated, our shallow MLP structure dynamic model cannot fully capture the transition info leading to high transition model error, which result in the performance deficit. Our theoretical analysis in Sec. 5.3 supports this finding, and we will explore the efficacy of different dynamic model structures for future works. Generally, **our method shows competitive performance with the baselines in the deterministic environments.**

## 7    CONCLUSION

In this paper, we presented a novel model-enhanced adversarial inverse reinforcement learning framework starting from Maximum Causal Entropy framework by incorporating model-based techniques with reward shaping, specifically designed to enhance performance in stochastic environments with significant sample efficiency improvement comparing to existing approaches and maintain competitive performance in deterministic setting. The theoretical analysis provides guarantees on the optimal policy invariance under the transition model involved reward shaping and highlight the relationship between performance gap and transition model error, showing that the gaps becomes negligible with a well-learned model. Empirical evaluations on Mujoco benchmark environments validate the effectiveness of our method, showcasing its superior performance and sample efficiency across different tasks. Future works will focus on further refining the model estimation process to handle more complex and dynamic environments and exploring extensions of the framework to multi-agent and hierarchical reinforcement learning scenarios. Additionally, it would be valuable to investigate the generalization ability of our framework in a transfer learning tasks. Overall, our approach offers a promising direction for advancing model-based adversarial IRL, with the potential to scale to a broader range of real-world applications.

## 8    REPRODUCIBLE STATEMENT

This work uses the open-source MuJoCo (Todorov et al., 2012) as the benchmark. The practical implementation of our method is built on the `CleanRL` repository (Huang et al., 2022). All hyperparameters to reproduce our experimental results, including learning rates and transition model settings, are explicitly listed in Appendix F. For every reported result, we averaged the performance over three random seeds, and the seed initialization is included for exact reproducibility.

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

## A    REWARD SHAPING SOFT OPTIMAL POLICY

**Theorem A.1.** *Let $R$ and $\hat{R}$ be two reward functions. $R$ and $\hat{R}$ induce the same soft optimal policy under all transition dynamics $\mathcal{T}$ if $\hat{R}(s_t, a_t) = R(s_t, a_t) + \gamma\mathbb{E}_{\mathcal{T}}[\phi(s_{t+1})|s_t, a_t] - \phi(s_t)$ for some potential-shaping function $\phi : \mathcal{S} \to \mathbb{R}$.*

*Proof.* According to Soft VI ( Eq. (2)), we can expand the representation of $Q_{\hat{R}}^{soft}(s_t, a_t)$ as follows.

$$Q_{\hat{R}}^{soft}(s_t, a_t) = R(s_t, a_t) + \gamma\mathbb{E}_{\mathcal{T}}\left[\phi(s_{t+1})|s_t, a_t\right] - \phi(s_t) + \gamma\mathbb{E}_{\mathcal{T}}\left[V_{\hat{R}}^{soft}(s_{t+1})|s_t, a_t\right],$$

$$Q_{\hat{R}}^{soft}(s_t, a_t) + \phi(s_t) = R(s_t, a_t) + \gamma\mathbb{E}_{\mathcal{T}}\left[V_{\hat{R}}^{soft}(s_{t+1}) + \phi(s_{t+1})|s_t, a_t\right],$$

$$Q_{\hat{R}}^{soft}(s_t, a_t) + \phi(s_t) = R(s_t, a_t) + \gamma\mathbb{E}_{\mathcal{T}}\left[\log\sum_{a\in\mathcal{A}}\exp\left(Q_{\hat{R}}^{soft}(s_{t+1}, a) + \phi(s_{t+1})\right)|s_t, a_t\right].$$

From above induction, we can tell that $Q_{\hat{R}}^{soft}(s_t, a_t) + \phi(s_t)$ satisfy the soft bellmen update with original $R$. Thus, with simple induction, we can arrive that $Q_R^{soft}(s_t, a_t) = Q_{\hat{R}}^{soft}(s_t, a_t) + \phi(s_t)$. Then, we can derive the advantage function

$$\begin{aligned}
A_{\hat{R}}^{soft}(s_t, a_t) &= Q_{\hat{R}}^{soft}(s, a) - V_{\hat{R}}^{soft}(s_t) \\
&= Q_{\hat{R}}^{soft}(s_t, a_t) - \log\sum_{a\in\mathcal{A}}\exp\left(Q_{\hat{R}}^{soft}(s_t, a_t)\right) \\
&= Q_{\hat{R}}^{soft}(s_t, a_t) + \phi(s_t) - \log\sum_{a\in\mathcal{A}}\exp\left(Q_{\hat{R}}^{soft}(s_t, a_t) + \phi(s_t)\right) \\
&= Q_R^{soft}(s_t, a_t) - \log\sum_{a\in\mathcal{A}}\exp\left(Q_R^{soft}(s_t, a_t)\right) \\
&= A_R^{soft}(s_t, a_t).
\end{aligned}$$

$\square$

## B    ADVERSARIAL REWARD LEARNING

**Proposition B.1.** *Consider an undiscounted MDP. Suppose $f_\theta$ and $\pi$ at current iteration are the soft-optimal advantage function and policy for reward function $R_\theta$. Minimising the cross-entropy loss of the discriminator under generator $\pi$ is equivalent to maximising the log-likelihood under Maximum Causal Entropy IRL.*

*Proof.*

$$\begin{aligned}
\mathcal{L}_{IRL}(\mathcal{D}_{exp}, \theta) &= \mathbb{E}_{\mathcal{D}_{exp}}[\log p_\theta(\tau)] \\
&= \mathbb{E}_{\mathcal{D}_{exp}}\left[\sum_{t=0}^{T-1}\log\pi(a_t|s_t) + \log\rho_0 + \sum_{t=1}^{T}\log\mathcal{T}(s_{t+1}|s_t, a_t)\right] \\
&= \mathbb{E}_{\mathcal{D}_{exp}}\left[\sum_{t=0}^{T-1}\left(Q_\theta^{soft}(s_t, a_t) - V_\theta^{soft}(s_t)\right)\right] + \text{constant}.
\end{aligned}$$

Breaking down above equations with soft VI (Eq. (2)), we can arrive the following.

$$\mathbb{E}_{\mathcal{D}_{exp}}\left[\sum_{t=0}^{T-1}R_\theta(s_t, a_t)\right] + \mathbb{E}_{\mathcal{D}_{exp}}\left[\sum_{t=0}^{T-2}\mathbb{E}_{\mathcal{T}}\left[V_\theta^{soft}(s_{t+1})|s_t, a_t\right]\right] - \mathbb{E}_{\mathcal{D}_{exp}}\left[\sum_{t=0}^{T-1}V_\theta^{soft}(s_t)\right]. \quad (13)$$

Next we will derive the gradient of the loss.

$$\nabla_\theta \mathcal{L}(\mathcal{D}_{exp}, \theta) = \underbrace{\nabla_\theta \mathbb{E}_{\mathcal{D}_{exp}} \left[ \sum_{t=0}^{T-1} R_\theta(s_t, a_t) \right]}_{A} +$$

$$\underbrace{\nabla_\theta \mathbb{E}_{\mathcal{D}_{exp}} \left[ \sum_{t=0}^{T-2} \left( \mathbb{E}_{\mathcal{T}} \left[ V_\theta^{soft}(s_{t+1}) | s_t, a_t \right] \right) - V_\theta^{soft}(s_{t+1}) \right]}_{B} - \underbrace{\nabla_\theta V_\theta^{soft}(s_0)}_{C}. \quad (14)$$

Let's get explicit expression of each part.

$$A = \mathbb{E}_{\mathcal{D}_{exp}} \left[ \sum_{t=0}^{T-1} \nabla_\theta R_\theta(s_t, a_t) \right]$$

$$C = \nabla_\theta \log \sum_{a_t \in \mathcal{A}} \exp Q_\theta^{soft}(s_t, a_t)$$

$$= \sum_{a_t \in \mathcal{A}} \pi(a_t | s_t) \nabla_\theta Q_\theta^{soft}(s_t, a_t)$$

$$= \mathbb{E}_\pi \left[ \sum_{t=0}^{T-1} \nabla_\theta R_\theta(s_t, a_t) \right].$$

In our case, the transition function $\mathcal{T}$ is estimated by an approximation function $\hat{\mathcal{T}}$, which is updated with samples from $\mathcal{D}_{exp}$ and samples from off-policy buffer $\mathcal{D}_{env}$, thus we can safely drop the $\mathbb{E}_{\mathcal{T}}$ here. And $B$ term will cancel out, ending up to $0$. To summarize, the gradient of log MLE loss of MCE IRL is the following.

$$\nabla_\theta \mathcal{L}_{IRL}(\mathcal{D}_{exp}, \theta) = \mathbb{E}_{\mathcal{D}_{exp}} \left[ \sum_{t=0}^{T-1} \nabla_\theta R_\theta(s_t, a_t) \right] - \mathbb{E}_\pi \left[ \sum_{t=0}^{T-1} \nabla_\theta R_\theta(s_t, a_t) \right]. \quad (15)$$

Next, we will start to derive the gradient of cross-entropy discriminator training loss. Remember the discriminator loss is defined in Eq 9.

$$\log D_\theta(s, a, \mathcal{T}) = f_\theta(s, a, \mathcal{T}) - \log(\exp\{f_\theta(s, a, \mathcal{T})\} + \pi(a|s)),$$
$$\log(1 - D_\theta(s, a, \mathcal{T})) = \log \pi(a|s) - \log(\exp\{f_\theta(s, a, \mathcal{T})\} + \pi(a|s)).$$

Then, the gradient of each term is as follow:

$$\nabla_\theta \log D_\theta(s, a, \mathcal{T}) = \nabla_\theta f_\theta(s, a, \mathcal{T}) - \frac{\exp\{f_\theta(s, a, \mathcal{T})\} \nabla_\theta f_\theta(s, a, \mathcal{T})}{\exp\{f_\theta(s, a, \mathcal{T})\} + \pi(a|s)},$$

$$\nabla_\theta \log(1 - D_\theta(s, a, \mathcal{T})) = -\frac{\exp\{f_\theta(s, a, \mathcal{T})\} \nabla_\theta f_\theta(s, a, \mathcal{T})}{\exp\{f_\theta(s, a, \mathcal{T})\} + \pi(a|s)}.$$

Since $\pi$ is trained by using $f_\theta$ as shaped reward, from soft VI we can derive that $\pi_{f_\theta}^*(a|s) = \exp A_{f_\theta}^{soft}(s, a)$. By assumption, we assume that $f_\theta$ is the advantage function of $R_\theta$, $f_\theta(s, a) = A_{R_\theta}^{soft}(s, a)$. From Thm 4.2, we know that $A_{R_\theta}^{soft}(s, a) = A_{f_\theta}^{soft}(s, a)$, which also implies that $\pi_{f_\theta}^* = \pi_{R_\theta}^*$. Then, we can deduce the gradient of the loss of discriminator.

$$-\nabla_\theta \mathcal{L}_{disc} = \mathbb{E}_{\mathcal{D}_{exp}} \left[ \nabla_\theta \log D_\theta(s, a, \mathcal{T}) \right] + \mathbb{E}_\pi \left[ \nabla_\theta \log(1 - D_\theta(s, a, \mathcal{T})) \right]$$

$$= \mathbb{E}_{\mathcal{D}_{exp}} \left[ \frac{1}{2} \nabla_\theta f_\theta(s, a, \mathcal{T}) \right] - \mathbb{E}_\pi \left[ \frac{1}{2} \nabla_\theta f_\theta(s, a, \mathcal{T}) \right],$$

$$-2\nabla_\theta \mathcal{L}_{disc} = \mathbb{E}_{\mathcal{D}_{exp}} \left[ \nabla_\theta f_\theta(s, a, \mathcal{T}) \right] - \mathbb{E}_\pi \left[ \nabla_\theta f_\theta(s, a, \mathcal{T}) \right].$$

$$\square$$

## C   PERFORMANCE GAP ANALYSIS

**Lemma C.1** (Implicit Feasible Reward Set (Ng et al., 2000))**.** *Let $\mathfrak{B} = (\mathcal{M}', \pi^*)$ be an IRL problem. Then $R \in \mathcal{R}_{\mathfrak{B}}$ if and only if for all $(s, a) \in \mathcal{S} \times \mathcal{A}$ the following holds:*

$$Q_{\mathcal{M}' \cup R}^{\pi^*}(s, a) - V_{\mathcal{M}' \cup R}^{\pi^*}(s) = 0 \quad if \ \pi^*(a|s) > 0,$$
$$Q_{\mathcal{M}' \cup R}^{\pi^*}(s, a) - V_{\mathcal{M}' \cup R}^{\pi^*}(s) \leq 0 \quad if \ \pi^*(a|s) = 0.$$

Combined with the traditional Value Iteration of RL, we can write out the explicit form of the reward function $R$.

**Lemma C.2** (Explicit Feasible Reward Function (Metelli et al., 2021))**.** *With the above lemma conditions, $R \in \mathcal{R}_{\mathfrak{B}}$ if and only if there exist $\xi \in \mathbb{R}_{\geq 0}^{\mathcal{S} \times \mathcal{A}}$ and value function $V \in \mathbb{R}^{\mathcal{S}}$ such that:*

$$R(s, a) = V(s) - \gamma \sum_{s' \in \mathcal{S}} \mathcal{T}(s'|s, a) V(s') - \xi(s, a) \mathbb{I}\{\pi^*(a|s) = 0\}. \tag{16}$$

With Eq. (16), we can derive the following error bound between $R \in \mathcal{R}_{\mathfrak{B}}^E$ and $\hat{R}^E \in \mathcal{R}_{\hat{\mathfrak{B}}}$.

**Theorem C.3** (Reward Function Error Bound)**.** *Let $\mathfrak{B} = (\mathcal{M}', \pi^*)$ and $\hat{\mathfrak{B}} = (\hat{\mathcal{M}}', \pi^*)$ be two IRL problems, then for any $R^E \in \mathcal{R}_{\mathfrak{B}}$ there is a corresponding $\hat{R}^E \in \mathcal{R}_{\hat{\mathfrak{B}}}$ such that*

$$\|R^E - \hat{R}^E\| \leq \frac{\gamma}{1 - \gamma} |\mathcal{S}| \epsilon_{\mathcal{T}} R_{\max}. \tag{17}$$

*Proof.* From Lem. C.2, we can derive the following representations of $R$ and $\hat{R}$ with the same set of $V$ and $\xi$:

$$R^E(s, a) = V(s) - \gamma \sum_{s' \in \mathcal{S}} \mathcal{T}(s'|s, a) V(s') - \xi(s, a) \mathbb{I}\{\pi^*(a|s) = 0\},$$
$$\hat{R}^E(s, a) = V(s) - \gamma \sum_{s' \in \mathcal{S}} \hat{\mathcal{T}}(s'|s, a) V(s') - \xi(s, a) \mathbb{I}\{\pi^*(a|s) = 0\}.$$

The difference between $R^E$ and $\hat{R}^E$ can be bounded as follows:

$$\|R^E - \hat{R}^E\| \leq \gamma \sum_{s' \in \mathcal{S}} D_{TV}\left(\mathcal{T}(s'|s, a) | \hat{\mathcal{T}}(s'|s, a)\right) \cdot \|V(s')\|.$$

Given that the total variation distance between the two dynamics is bounded by $\epsilon_{\mathcal{T}}$, and the reward function is bounded by $R_{\max}$, together with the definition of the value function, we have $\|V\|_\infty \leq \frac{R_{\max}}{1-\gamma}$. Substituting these bounds, we derive the following inequality:

$$\|R^E - \hat{R}^E\| \leq \frac{\gamma}{1 - \gamma} |\mathcal{S}| \epsilon_{\mathcal{T}} R_{\max}.$$

$\square$

Next, we will propagate this bound to the value functions of optimal policy regarding different reward functions $R^E$ and $\hat{R}^E$. From the traditional Value iteration, we can write out the value function.

$$V_{\mathcal{M}' \cup R^E}^{\pi}(s) = \sum_{a \in \mathcal{A}} \pi(a|s) \sum_{s' \in \mathcal{S}} \mathcal{T}(s'|s, a) \left[R^E(s, a) + \gamma V_{\mathcal{M}' \cup R^E}^{\pi}(s')\right]. \tag{18}$$

**Lemma C.4** (Value Function Error under same policy and different rewards and MDP)**.** $\|V_{\mathcal{M}' \cup R}^{\pi}(s) - V_{\hat{\mathcal{M}}' \cup \hat{R}}^{\pi}(s)\|$*: the performance difference of the same policy in different MDPs.*

$$\|V_{\mathcal{M}' \cup R^E}^{\pi}(s) - V_{\hat{\mathcal{M}}' \cup \hat{R}^E}^{\pi}(s)\| \leq \epsilon_{\mathcal{T}} \frac{1 + \gamma}{(1 - \gamma)^2} R_{\max} |\mathcal{S}|. \tag{19}$$

*Proof.*

$$||V^\pi_{\mathcal{M}'\cup R^E}(s) - V^\pi_{\hat{\mathcal{M}}'\cup\hat{R}^E}(s)||$$

$$\leq \sum_{a\in\mathcal{A}}\pi(a|s)\sum_{s'\in\mathcal{S}}||\mathcal{T}(s'|s,a)\left[R^E(s,a)+\gamma V^\pi_{\mathcal{M}'\cup R^E}(s')\right] - \hat{\mathcal{T}}(s'|s,a)\left[R^E(s,a)+\gamma V^\pi_{\mathcal{M}'\cup R^E}(s')\right]$$

$$+ \hat{\mathcal{T}}(s'|s,a)\left[R^E(s,a)+\gamma V^\pi_{\mathcal{M}'\cup R^E}(s')\right] - \hat{\mathcal{T}}(s'|s,a)\left[\hat{R}^E(s,a)+\gamma V^\pi_{\hat{\mathcal{M}}'\cup\hat{R}^E}(s')\right]||$$

$$\leq \sum_{a\in\mathcal{A}}\pi(a|s)\sum_{s'\in\mathcal{S}}(\epsilon_\mathcal{T}\frac{R_{\max}}{1-\gamma} + \hat{\mathcal{T}}(s'|s,a)\epsilon_\mathcal{T}(\frac{\gamma R_{\max}}{1-\gamma}|\mathcal{S}| + \gamma||V^\pi_{\mathcal{M}'\cup R^E}(s') - V^\pi_{\hat{\mathcal{M}}'\cup\hat{R}^E}(s')||))$$

$$= \sum_{a\in\mathcal{A}}\pi(a|s)(\epsilon_\mathcal{T}\frac{R_{\max}}{1-\gamma}|\mathcal{S}| + \epsilon_\mathcal{T}\frac{\gamma R_{\max}}{1-\gamma}|\mathcal{S}| + \gamma||V^\pi_{\mathcal{M}'\cup R^E}(s') - V^\pi_{\hat{\mathcal{M}}'\cup\hat{R}^E}(s')||)$$

$$\leq \epsilon_\mathcal{T}\frac{1+\gamma}{1-\gamma}R_{\max}|\mathcal{S}| + \gamma||V^\pi_{\mathcal{M}'\cup R^E}(s') - V^\pi_{\hat{\mathcal{M}}'\cup\hat{R}^E}(s')||$$

$$\leq \epsilon_\mathcal{T}\frac{1+\gamma}{(1-\gamma)^2}R_{\max}|\mathcal{S}|.$$

$$(20)$$

$\square$

**Lemma C.5.** *Let $||V^{\pi_1}_{\hat{\mathcal{M}}'\cup\hat{R}^E}(s) - V^{\pi_2}_{\hat{\mathcal{M}}'\cup\hat{R}^E}(s)||$ denote the performance difference between different policies $\pi_1$ and $\pi_2$ in the same learned MDP ([Viano et al., 2021](); [Zhang et al., 2020]()). The following inequality holds:*

$$||V^{\pi_1}_{\hat{\mathcal{M}}'\cup\hat{R}^E}(s) - V^{\pi_2}_{\hat{\mathcal{M}}'\cup\hat{R}^E}(s)|| \leq \frac{\gamma}{(1-\gamma)^2}\epsilon_\mathcal{T}R_{\max}.$$

**Theorem C.6** (Performance Difference Bound). *The performance difference between the optimal policies ($\pi^*$ and $\hat{\pi}^*$) in corresponding MDPs ($\mathcal{M}'\cup R^E$ and $\hat{\mathcal{M}}'\cup\hat{R}^E$) can be bounded as follows:*

$$||V^{\pi^*}_{\mathcal{M}'\cup R^E} - V^{\hat{\pi}^*}_{\hat{\mathcal{M}}'\cup\hat{R}^E}|| \leq \epsilon_\mathcal{T}\left[\frac{\gamma}{(1-\gamma)^2}R_{\max} + \frac{1+\gamma}{(1-\gamma)^2}R_{\max}|\mathcal{S}|\right].$$

$$(21)$$

*Proof.*

$$||V^{\pi^*}_{\mathcal{M}'\cup R^E}(s) - V^{\hat{\pi}^*}_{\hat{\mathcal{M}}'\cup\hat{R}^E}(s)||$$

$$\leq ||V^{\pi^*}_{\hat{\mathcal{M}}'\cup\hat{R}^E}(s) - V^{\hat{\pi}^*}_{\hat{\mathcal{M}}'\cup\hat{R}^E}(s)|| + ||V^{\hat{\pi}^*}_{\mathcal{M}'\cup R^E}(s) - V^{\hat{\pi}^*}_{\hat{\mathcal{M}}'\cup\hat{R}^E}(s)||$$

$$= \epsilon_\mathcal{T}\frac{\gamma}{(1-\gamma)^2}R_{\max} + \epsilon_\mathcal{T}\frac{1+\gamma}{(1-\gamma)^2}R_{\max}|\mathcal{S}|$$

$$= \epsilon_\mathcal{T}\left[\frac{\gamma}{(1-\gamma)^2}R_{\max} + \frac{1+\gamma}{(1-\gamma)^2}R_{\max}|\mathcal{S}|\right].$$

$\square$

# D    GRAPH RESULTS

Below is the testing return diagrams from stochastic Mujoco Environments under `1M` landscape.

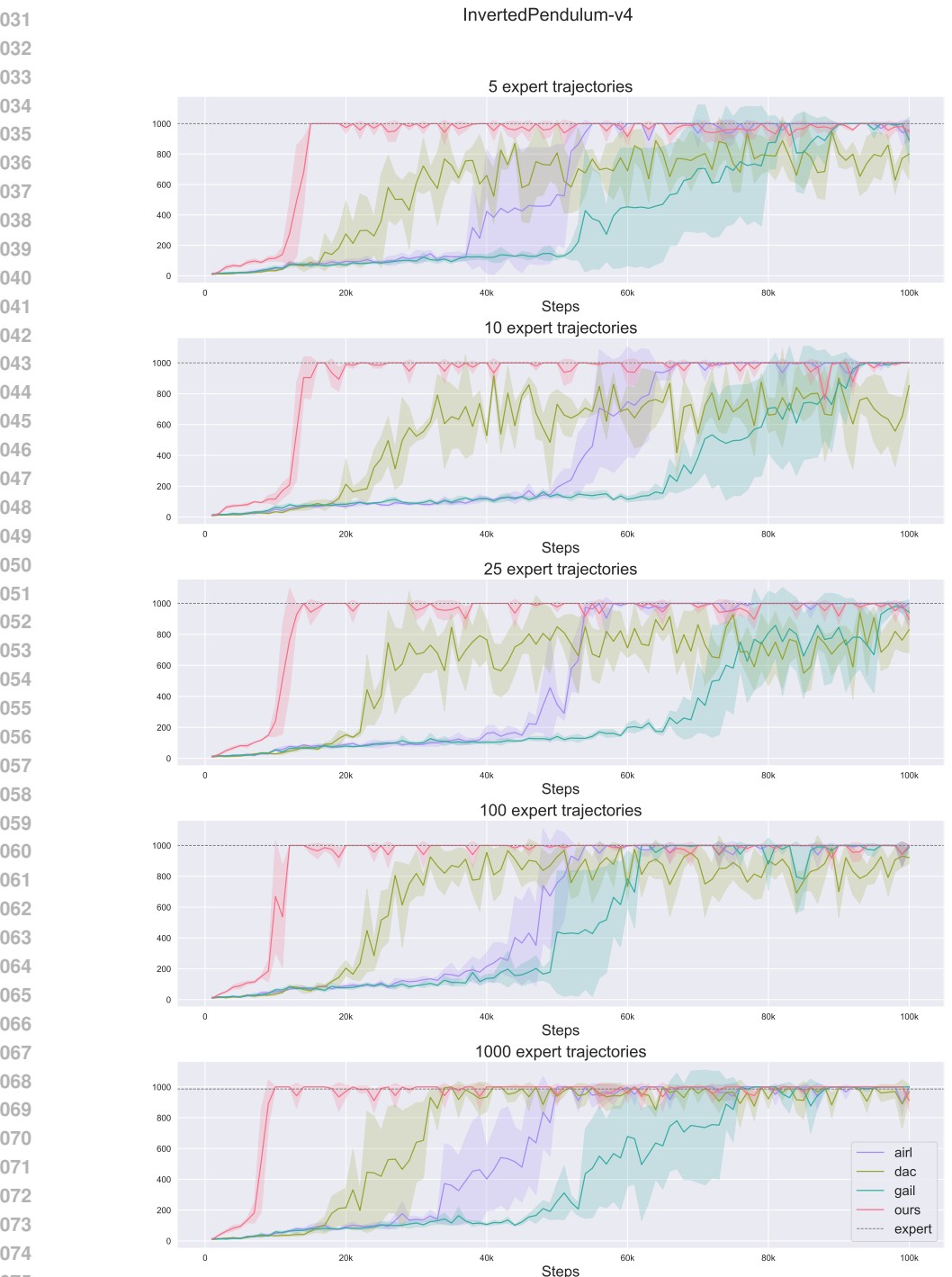

Figure 3: Training return diagram averaging across three seeds for different numbers of expert trajectories in Stochastic `InvertedPendulum-v4`.

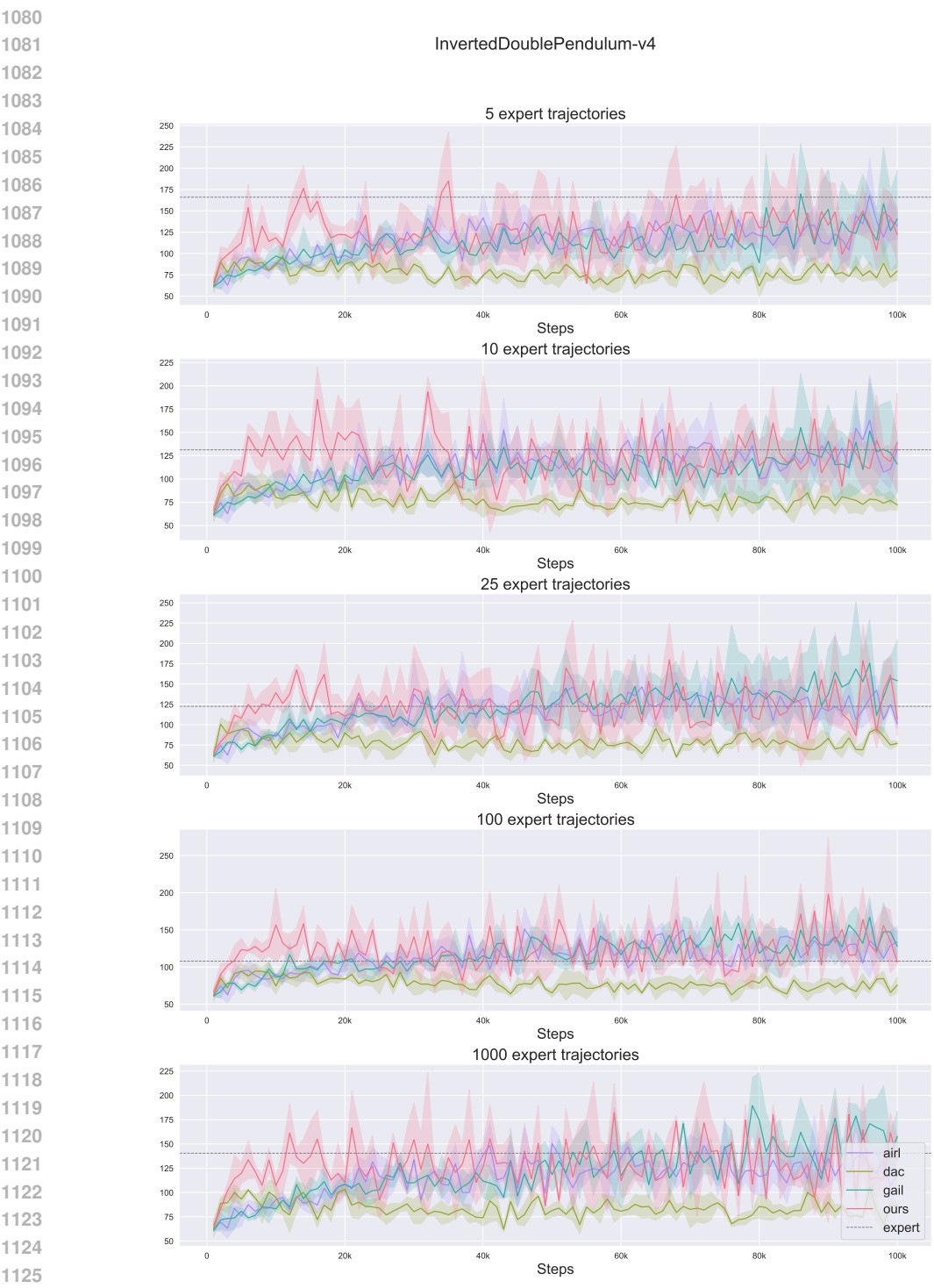

Figure 4: Training return diagram averaging across three seeds for different numbers of expert trajectories in Stochastic `InvertedDoublePendulum-v4`.

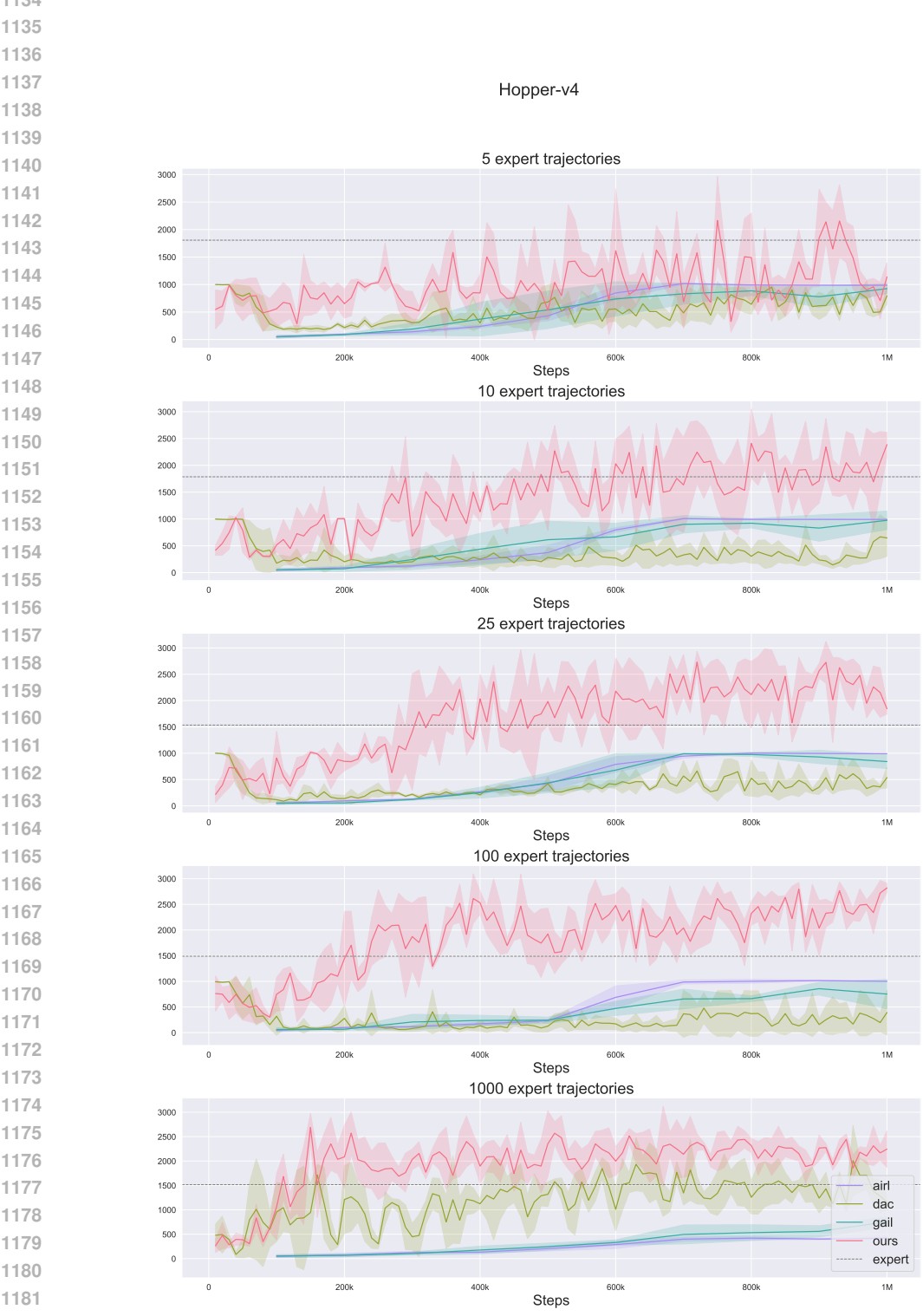

Figure 5: Training return diagram averaging across three seeds for different numbers of expert trajectories in Stochastic `Hopper-v4`.

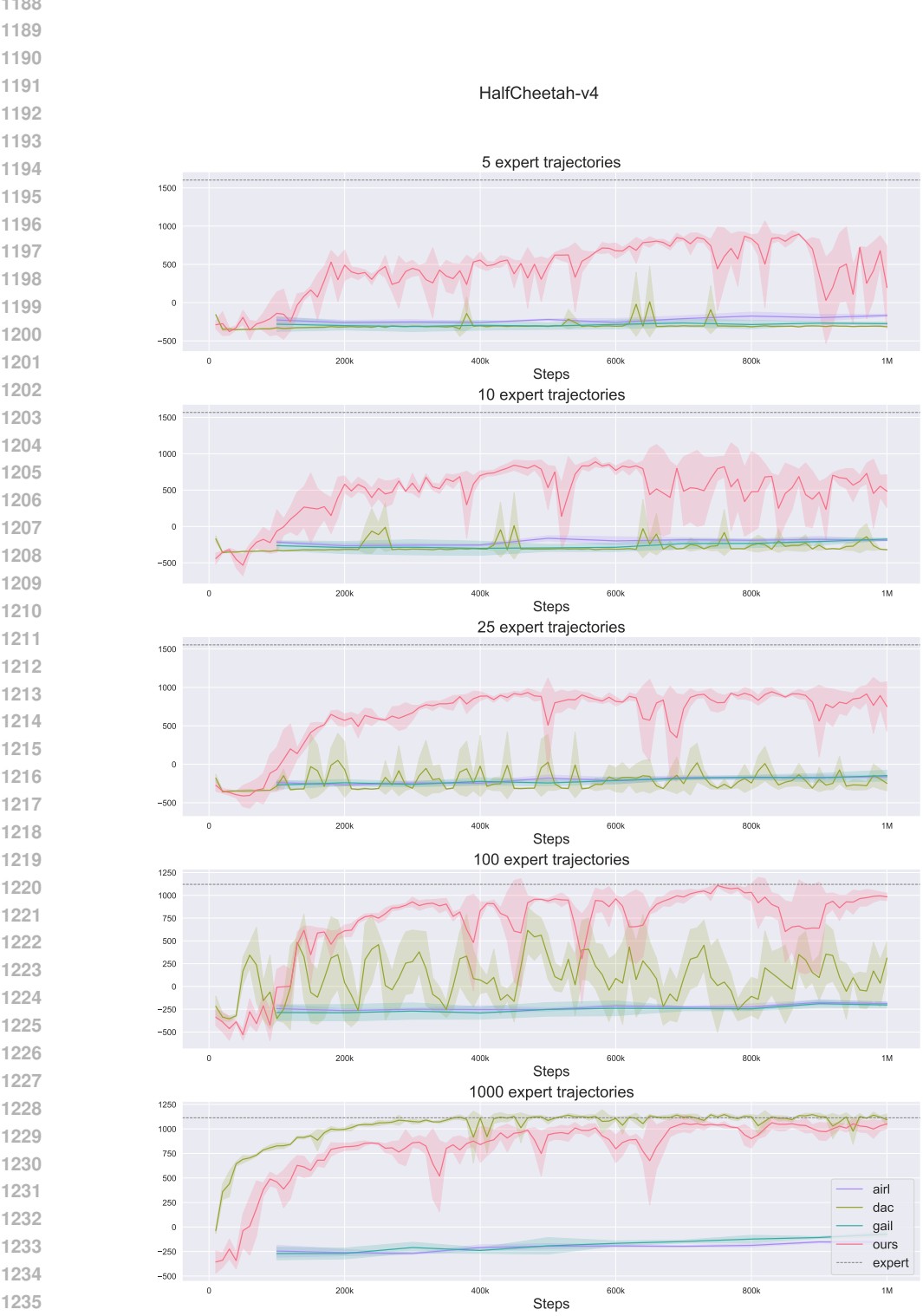

Figure 6: Training return diagram averaging across three seeds for different numbers of expert trajectories in Stochastic `HalfCheetah-v4`.

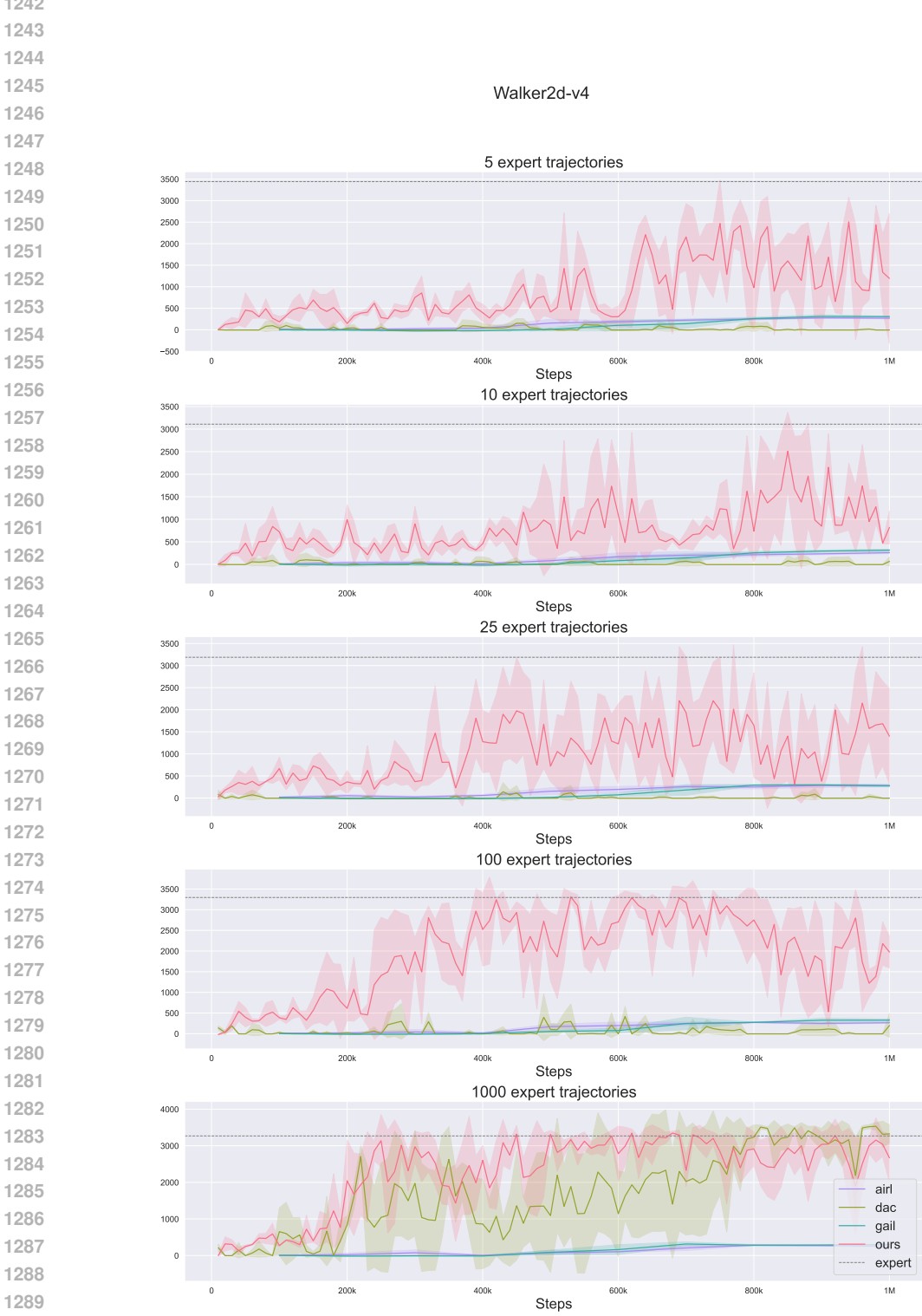

Figure 7: Training return diagram averaging across three seeds for different numbers of expert trajectories in Stochsatic `Walker2d-v4`.

# E ABLATION STUDY

## E.1 ROBUSTNESS TO STOCHASTICITY

In this study, we examine the robustness of our method across varying levels of stochasticity in the environment. Following the same setup as in our main experiments, we introduce an unknown Gaussian noise with different standard deviations in `InvertedPendulum-v4` to simulate increased stochasticity. As shown in Appendix E.1 and Fig. 8, our method consistently recovers expert-level performance despite the presence of stochastic disturbances. However, as the level of stochasticity increases, we observe that training stability decreases, as reflected in the increased variance in Fig. 8.

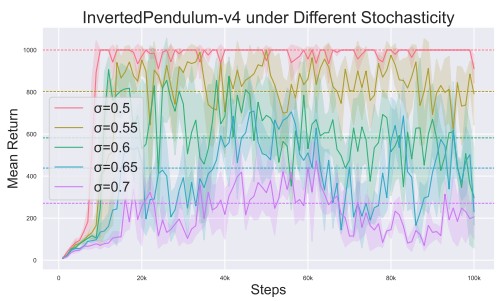

| Std | Expert | Ours |
|---|---|---|
| 0.5 | $1000.0_{\pm 0.0}$ | $1000.0_{\pm 0.0}$ |
| 0.55 | $802.4_{\pm 305.8}$ | $1000.0_{\pm 0.0}$ |
| 0.6 | $582.1_{\pm 360.5}$ | $906.3_{\pm 59.1}$ |
| 0.65 | $438.2_{\pm 322.3}$ | $709.9_{\pm 80.6}$ |
| 0.7 | $270.7_{\pm 236.0}$ | $472.7_{\pm 85.7}$ |

Table 4: Best performance of expert and our method in `InvertedPendulum-v4` environments with different Gaussian noises (standard deviations ranging from 0.5 to 0.7) for stochasticity under provided 100 expert trajectories.

Figure 8: Training return diagram averaging across three seeds for different numbers of expert trajectories in `InvertedPendulum-v4`.

## E.2 MODEL ESTIMATION ERROR AND REWARD LEARNING

In this study, we empirically evaluate the effect of dynamic model learning errors on our method's performance, extending the theoretical analysis presented in Sec. 5.3. To isolate the impact of model errors specifically on reward learning, we use SAC on real trajectories for policy optimization, thereby removing any influence of model errors on trajectory generation that would typically affect model-based policy optimization. To quantify the relationship between model errors and performance, we standardize the model architecture as a 2-layer MLP with varying hidden layer dimensions from 8 to 256 to adjust model capacity. Our experiments are conducted in `HalfCheetah-v4` with random, policy-unknown Gaussian noise (mean 0 and standard deviation 0.5), as described in Sec. 6. From Fig. 10 and Fig. 9, we observe the general trend that as modeling error decrease together with increasing capacity of the model structure, performances also increases, which is obvious when hidden dimension bumps up from 8 to 16 and 16 to 32. As transition model error narrows down, the performance improvement also becomes less obvious.

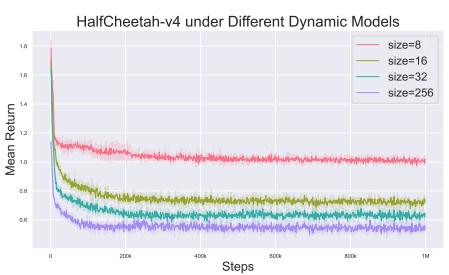

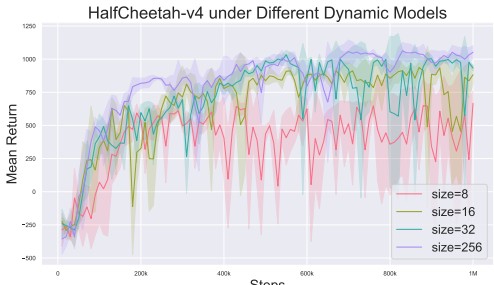

Figure 9: Transition model learning error diagram averaging across three seeds for 10 expert trajectories in `HalfCheetah-v4`.

Figure 10: Training return diagram averaging across three seeds for 10 expert trajectories in `HalfCheetah-v4`.

### E.3 DOES MODEL-BASED TRAJECTORIES GENERATION HELP?

In this study, we empirically investigate the effectiveness of model-based policy optimization on our model-enhanced reward shaping IRL framework. We compare three off-policy approaches namely Discriminator Actor-Critic (DAC (Kostrikov et al., 2018)), model-enhanced reward shaping with pure SAC for policy optimization (labeled as $mbirl\_sac$), and our original model-enhanced reward shaping with model-based technique for policy optimization. Noted that synthetic data is also not used in reward learning in $mbirl\_sac$ approach. We conduct the experiment in stochastic Hopper-v4 with 1000 provided expert trajectories. From Fig. 11 and Appendix E.3, we can tell that both methods using model-enhanced reward shaping have much better performance and sample efficiency comparing to DAC which doesn't have. In terms of performance, both methods perform at the similar level. However, as the synthetic trajectories generation boost the training process, our model-based method has better sample efficiency than the pure SAC-based method.

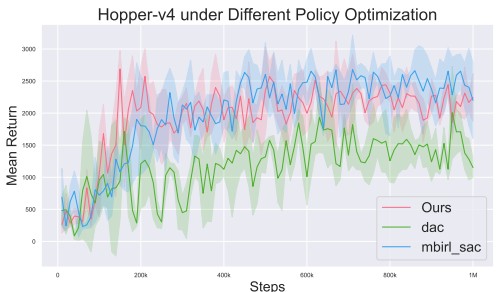

Figure 11: Performance diagram averaging across three seeds for different algorithms in Hopper-v4 with 1000 expert trajectories provided. DAC is in red color; $mbirl\_sac$ is in green; Our method is in blue

| Method | Performance |
|--------|-------------|
| DAC | $2007.1_{\pm 719.7}$ |
| $mbirl\_sac$ | $2694.5_{\pm 77.5}$ |
| Ours | $2798.8_{\pm 82.9}$ |

Table 5: Best performance of three methods in stochastic Hopper-v4 environment with under provided 1000 expert trajectories.

## F  IMPLEMENTATION DETAILS

For our framework, we use two identical 2-layer Multi-Layer Perceptrons (MLPs) with 100 hidden units and ReLU activations for both the reward function $R$ and the shaping potential function $\phi$. To initialize the replay buffer for both **DAC** and ours, we collect 1,000 steps samples in `InvertedPendulum-v4` and `InvertedDoublePendulum-v4`, and 10,000 steps samples in `Hopper-v4`, `HalfCheetah-v4`, and `Walker2d-v4` with initial policy. During this pre-training phase, we also update the transition model at each step to mitigate divergence might happen at the beginning of the training. Additionally, the transition model is only trained using samples from real environment buffer $\mathcal{D}_{env}$ in policy optimization section before actor and critics updates during training phase. As discussed in Sec. 5, the size of the synthetic data buffer $\mathcal{D}_{gen}$ and the ratio of samples drawn from it increase as the model accuracy improves. Both parameters increase linearly with training steps, up to a maximum synthetic-to-real data ratio of 0.5 per training step and a maximum buffer size of 1 million samples in $\mathcal{D}_{gen}$. For consistency in comparisons, we used similar network structures and hyper-parameters for **AIRL, GAIL, and DAC** baselines, which we reference the implementations from Arulkumaran & Lillrank (2024) and Gleave et al. (2022). Detailed hyper-parameters for these networks are provided in the table below. For on-policy baselines **AIRL** and **GAIL**, the rollout length is set to 1,000 for `InvertedPendulum-v4` and `InvertedDoublePendulum-v4`, and 5,000 for `Hopper-v4`, `Walker2d-v4`, and `HalfCheetah-v4`. For the SAC and PPO policy optimization components, we reference implementations from the `CleanRL` repository (Huang et al., 2022). The code for our method and all baseline implementations can be found here: https://anonymous.4open.science/r/MBIRL-4C2F/README.md.

Table 6: Hyper-parameters table.

| Hyper-parameter | Value |
| --- | --- |
| Seeds | 0, 5, 10 |
| Buffer Size | 1M |
| Batch Size | 128 |
| Max Grad Norm | 10 |
| Starting Steps | 1,000/10,000 |
| Global Timesteps | 100k/1M |
| Discount Factor | 0.99 |
| Model-based Policy Optimization | |
| Learning Rate for Actor | 3e-4 |
| Learning Rate for Critic | 3e-4 |
| Learning Rate for Model | 3e-4 |
| Network Layers | 3 |
| Policy Network Neurons | [64, 64] |
| Critic Network Neurons | [128, 128] |
| Model Network Neurons | [256, 256] |
| Activation | Tanh(Policy)/ReLU |
| Optimizer | Adam |
| Initial Entropy | $-|\mathcal{A}|$ |
| Learning Rate for Entropy | 3e-4 |
| Train Frequency for Actor | 1 |
| Train Frequency for Critic | 1 |
| Train Frequency for Model | 1 |
| Synthetic and Real Data Mix Coef | 0.5 |
| Horizon($H$) | 2 |
| Adversarial Discriminator | |
| Learning Rate | 3e-4 |
| $R$ Network Neurons | [100, 100] |
| $\phi$ Network Neurons | [100, 100] |
| Optimizer | Adam |
| Loss | Binary Cross-Entropy |

