# OpenReview forum: "Model-Enhanced Adversarial Inverse Reinforcement Learning with Model Estimation Reward Shaping in Stochastic Environments"
_ICLR.cc/2025/Conference — Submitted to ICLR 2025_

### Official Review · Reviewer_caVP · 2024-10-20

**Soundness:** 3
**Presentation:** 3
**Contribution:** 3
**Rating:** 5
**Confidence:** 4

**Summary:**

To address the issue of poor performance of traditional adversarial inverse reinforcement learning (AIRL) in stochastic environments, this paper incorporates environmental information into the reward model training and propose a new AIRL algorithm, which effectively improves performance in stochastic environments.

**Strengths:**

1) Writing: The article is overall very well-written, with clear structure and logical flow, rigorous theoretical derivations, and sufficient experiments.

2) Algorithm: The approach is innovative. By integrating environmental information into the reward model learning process, the method effectively enhances the algorithm’s performance in stochastic environments.

3) Theory: The paper derives that minimizing the discriminator’s cross-entropy loss is equivalent to maximizing the log-likelihood under MCE IRL.

**Weaknesses:**

1) Algorithm: Since the trained environment model differs from the real model, the generated trajectories are biased, which affects the learning of the reward model. Although the theoretical analysis in the paper discuss this, the algorithm design does not account for this issue.

2) Theory: The theoretical upper bound in the paper is too loose (i.e., the theoretical description is too broad) and not tightly integrated with the designed algorithm. The theoretical conclusions do not adequately reflect the properties of the proposed algorithm. Specifically:

i) The theoretical analysis of the reward model only considers the error of the learned environment model. However, the reward model is trained using both data generated by the environment model and data generated through interaction with the real environment. The reward model's error involves not only model errors but also data coverage, sampling errors, etc.

ii) The performance analysis only considers the reward model's error, but performance is also influenced by the RL algorithm itself. For example, the collected data is generated through interactions between the learned policy and the environment, and the learned policy is influenced by both the reward model and the RL algorithm.

3) Experiments: The experimental details are insufficient. Key parameters such as model rollout length, the proportion of model-generated data (and other parameters from Algorithm 1) are not provided. Additionally, details on the training of the environment model are lacking (e.g., is the environment model continuously updated during policy learning? On what data is each update based?).

**Questions:**

1) The proposed algorithm uses the learned policy to continuously interact with the real environment (or a realistic simulated environment) to collect data $\mathcal{D}_{env}$. However, if the real environment is already known, why is there a need to train an environment model and generate data? From my perspective, having both seems contradictory. Typically, when the real environment is unknown, current research uses collected offline data to train the environment model, which is then used to augment the dataset in order to improve sample efficiency.

2) Since the paper primarily focuses on reward model learning, the author should compare and analyze the gap between the learned reward model and the real reward model (this can be done by comparing the results after normalizing both models).

3) The quality of the data generated by the learned environment model is significantly influenced by the rollout length $H$. The author should provide a brief discussion on how $H$ is selected, and explain how the environment model is updated during the policy learning process.

---

> ### Author Response · Authors · 2024-11-18
> **Response to Reviewer caVP [1/2]**
>
> We sincerely appreciate reviewer caVP's recognition for our contribution and comments. And below is our response.
>
> ### **Weakness 1:**
> Since the trained environment model differs from the real model, the generated trajectories are biased, which affects the learning of the reward model. Although the theoretical analysis in the paper discuss this, the algorithm design does not account for this issue.
>
> **Answer:** In this work, the main contribution is to formulate transition model in the reward shaping to solve the MCE IRL problem under the stochastic setting. Improving transition model accuracy to mitigate distribution shift is not primary focus of our current work. Admittedly, the accuracy of transition model effect the performance reward learning, which we have extended an ablation study about this issue in Appendix E.2. We will explore different possibilities of transition model structures in future work.
>
> ### **Weakness 2:**
> * The theoretical upper bound in the paper is too loose (i.e., the theoretical description is too broad) and not tightly integrated with the designed algorithm. The theoretical conclusions do not adequately reflect the properties of the proposed algorithm
> * The theoretical analysis of the reward model only considers the error of the learned environment model. However, the reward model is trained using both data generated by the environment model and data generated through interaction with the real environment. The reward model's error involves not only model errors but also data coverage, sampling errors, etc.
> * The performance analysis only considers the reward model's error, but performance is also influenced by the RL algorithm itself. For example, the collected data is generated through interactions between the learned policy and the environment, and the learned policy is influenced by both the reward model and the RL algorithm.
>
> **Answer:**
> Our main contribution is proposing a model-based reward shaping to framework solve MCE IRL problem under stochastic setting. Consequently, the theoretical bound presented in our paper concentrates on providing a theoretical framework to connect possible suboptimality of recovered policy with transition model learning error. Thus, under this theoretical framework, there could be other possible algorithms with refinement or extension building upon our bound by using more guaranteed accurate transition model or considering convergence and sampling errors from other policy optimization methods. And we have empirically investigated the effect of different policy optimization methods under our model-enhanced reward shaping in ablation study Appendix E.3.
>
>
> ### **Weakness 3:**
> Experiments: The experimental details are insufficient. Key parameters such as model rollout length, the proportion of model-generated data (and other parameters from Algorithm 1) are not provided. Additionally, details on the training of the environment model are lacking (e.g., is the environment model continuously updated during policy learning? On what data is each update based?).
>
> **Answer:** We have provided more explicit implementation details in updated version. Please take a look at section 5.2 and Appendix F for details.

---

> > ### Author Response · Authors · 2024-11-18
> > **Response to Reviewer caVP [2/2]**
> >
> > ### **Question 1:**
> > The proposed algorithm uses the learned policy to continuously interact with the real environment (or a realistic simulated environment) to collect data
> > . However, if the real environment is already known, why is there a need to train an environment model and generate data? From my perspective, having both seems contradictory. Typically, when the real environment is unknown, current research uses collected offline data to train the environment model, which is then used to augment the dataset in order to improve sample efficiency.
> >
> > **Answer:** Our setting is the common setting in lots of IRL/MBIRL works[2,4,5,6,9,10], where only certain numbers of expert trajectories/behaviors are provided, and rewards and dynamics/simulated environments are unknown. The only distinction is that we assume the transition dynamic has stochastic nature instead of the deterministic one. Using offline expert data to train transition model has the following problems: 1, Offline expert trajectories have too limited data samples to learn dynamic model comparing to offline model-based RL approach[8]; 2, In our setting we only have offline expert trajectories, thus the learned stochastic transition model would be inevitably biased by the expert visited transition.
> >
> >
> > ### **Question 2:**
> > Since the paper primarily focuses on reward model learning, the author should compare and analyze the gap between the learned reward model and the real reward model (this can be done by comparing the results after normalizing both models).
> >
> > **Answer:** This paper is focusing on recovering expert policy through MCE IRL framework, which target to recover expert policy under stochastic setting with provided expert trajectories[2,3]. Unlike world model learning task to recover reward model, which is essentially a supervised learning task[11,12], IRL/IOC (Inverse Optimal Control) inherently suffers from an ill-posed nature [1,2], meaning that for only a given set of expert behaviors, there can exist many different rewards that explain their optimality, making the problem fundamentally challenging. Thus, for evaluation, we only focus on whether our method can recover expert policy using provided expert trajectories, which is same with literature work on IRL[2,3,4,13,14].
> >
> >
> > ### **Question 3:**
> > The quality of the data generated by the learned environment model is significantly influenced by the rollout length $H$. The author should provide a brief discussion on how $H$ is selected, and explain how the environment model is updated during the policy learning process.
> >
> > **Answer:** We have added the corresponding discussion about transition model training and hyper-parameters selection in the updated version. Please take a look at line 305-310 and Appendix F.
> >
> >
> > [1] Kalman, Rudolf Emil. "When is a linear control system optimal?." (1964): 51-60.
> >
> > [2]Arora, Saurabh, and Prashant Doshi. "A survey of inverse reinforcement learning: Challenges, methods and progress." Artificial Intelligence 297 (2021): 103500.
> >
> > [3]Ramachandran, Deepak, and Eyal Amir. "Bayesian Inverse Reinforcement Learning." IJCAI. Vol. 7. 2007.
> >
> > [4]Fu, Justin, Katie Luo, and Sergey Levine. "Learning robust rewards with adversarial inverse reinforcement learning." arXiv preprint arXiv:1710.11248 (2017).
> >
> > [5]Dexter, Gregory, Kevin Bello, and Jean Honorio. "Inverse reinforcement learning in a continuous state space with formal guarantees." Advances in Neural Information Processing Systems 34 (2021): 6972-6982.
> >
> > [6]Skalse, Joar, and Alessandro Abate. "Misspecification in inverse reinforcement learning." Proceedings of the AAAI Conference on Artificial Intelligence. Vol. 37. No. 12. 2023.
> >
> > [7]Michael Janner, Justin Fu, Marvin Zhang, and Sergey Levine. When to trust your model: Model-based policy optimization. Advances in neural information processing systems, 32, 2019.
> >
> > [8]Yu, Tianhe, et al. "Mopo: Model-based offline policy optimization." Advances in Neural Information Processing Systems 33 (2020): 14129-14142.
> >
> > [9]Jiankai Sun, Lantao Yu, Pinqian Dong, Bo Lu, and Bolei Zhou. Adversarial inverse reinforcement learning with self-attention dynamics model. IEEE Robotics and Automation Letters, 6(2):1880–1886, 2021.
> >
> > [10]Nir Baram, Oron Anschel, and Shie Mannor. Model-based adversarial imitation learning. arXivpreprint arXiv:1612.02179, 2016.
> >
> > [11]Nicklas Hansen, Xiaolong Wang, and Hao Su. Temporal difference learning for model predictive control. arXiv preprint arXiv:2203.04955, 2022.
> >
> > [12]Danijar Hafner, Timothy Lillicrap, Jimmy Ba, and Mohammad Norouzi. Dream to control: Learning behaviors by latent imagination. arXiv preprint arXiv:1912.01603, 2019.
> >
> > [13]Orsini, Manu, et al. "What matters for adversarial imitation learning?." Advances in Neural Information Processing Systems 34 (2021): 14656-14668.
> >
> > [14]Kostrikov, Ilya, et al. "Discriminator-actor-critic: Addressing sample inefficiency and reward bias in adversarial imitation learning." arXiv preprint arXiv:1809.02925 (2018).

---

> > > ### Author Response · Authors · 2024-11-22
> > > **Reminder to Review Rebuttal**
> > >
> > > Dear Reviewer caVP,
> > >
> > > We understand this is a busy time of year, but we wanted to kindly remind you to review our rebuttal. Your feedback is invaluable for us to improve our paper. Please feel free to let us know if there are any specific points that require further clarification.
> > >
> > > Thank you for your time and effort. We truly appreciate your contributions to this process!
> > >
> > > Best,
> > > Authors

---

> > > > ### Comment · Reviewer_caVP · 2024-11-25
> > > >
> > > > Thank you very much for the author's response and the detailed explanation of the experiments related to the transition model. However, some issues remain unresolved. e.g
> > > >
> > > > - why is it necessary to train an environment model when the real environment is already known?
> > > > - no comparison experiments between the learned reward model and the true reward model were provided.
> > > > - considering the theoretical shortcomings, there is a lack of in-depth analysis (not the main reason).
> > > >
> > > > Therefore, I maintain a negative stance.

---

> > > > > ### Author Response · Authors · 2024-11-25
> > > > >
> > > > > Thanks for your feedback.
> > > > >
> > > > > For your issue 1 and issue 2, we have already answered in Question 1 and Question 2. Specifically, for issue 1, the environment is unknown in our setting, which aligns with the extensive literature. For issue 2, it is not necessary to compare the learned reward model and the true reward model. The reason is that our objective is to learn a synthesis reward signal from the expert demonstration, instead of learning a true reward model.
> > > > >
> > > > > We'd love to address and explain any further concerns.

---

> > > > > > ### Comment · Reviewer_caVP · 2024-11-25
> > > > > >
> > > > > > Thanks for your reply.
> > > > > > - For issue 1, please author see algorithm 1 line 4 and 11. At each step of training, the author uses the currently learned strategy to interact with the real (simulated) environment to obtain data, and uses real data generated by real environment and model data generated by learned transition model to train the policy.
> > > > > > - For issue 2, there is indeed a discrepancy between the predicted values of the learned reward model and the true reward outputs, making direct comparison challenging. However, the trend predicted by the learned reward model should align with that of the true reward model. After normalization, the two should exhibit an approximately linear relationship.

---

> > > > > > > ### Author Response · Authors · 2024-11-25
> > > > > > >
> > > > > > > Thanks for prompt response!
> > > > > > >
> > > > > > > * For the first point, we are interacting with real environment to get only trajectories namely state-action-state transition pair. As for the dynamic of the real environment is totally unknown. Reviewer can think it as a blackbox system, each time we input action and observe output state. This is standard setting for online RL algorithms[1]
> > > > > > >
> > > > > > > * For the second part, the linear relationship doesn't necessarily hold. We can explain it through a simple discrete toy example. Imagine a robot navigating in a grid world. The expert demonstrations show the robot always moving towards a specific goal (e.g., the top-right corner of the grid and start at the lower-left corner). From these demonstrations, IRL aims to infer the underlying reward model that explains this behavior. Now consider two possible reward models:
> > > > > > >      - Reward Model A: The robot gets a positive reward (+10) for being at the goal and zero elsewhere
> > > > > > >      - Reward Model B: The robot gets a small reward (+1) for every step taken closer to the goal and a larger reward (+20) when it reaches the goal.
> > > > > > >     - Reward Model C: The robot gets a reward by how distance it moves away from the starting point.
> > > > > > >
> > > > > > > All models can perfectly explain the observed behavior, as the expert's actions (always moving towards the goal) maximize the total reward in either case. However, the absolute reward values differ between these models, and those reward mechanisms won't exhibit any kinds of linear relationship.
> > > > > > >
> > > > > > > We hope these explanations can address your concerns.
> > > > > > >
> > > > > > > [1]Sutton, Richard S., and Andrew G. Barto. Reinforcement learning: An introduction. MIT press, 2018.

---

> > > > > > > > ### Comment · Reviewer_caVP · 2024-11-26
> > > > > > > >
> > > > > > > > Thanks for your reply.
> > > > > > > > - Online RL usually does not require learning the transition model. The learned transition model is also a blackbox system, which inputs state-action pairs and outputs next state. The learned transition model is used normally in offline RL.
> > > > > > > > - I understand what the author means. I did not suggest that the author directly compare the difference between the two, but the comparison after normalization. The author can further consider this in the future.
> > > > > > > >
> > > > > > > > Therefore, considering other reviewers' comments, I also keep my score.

---

> > > > > ### Author Response · Authors · 2024-11-25
> > > > >
> > > > > We thank for Reviewer caVP getting back to us. Regarding all the three points pointed out by the reviewer, we have already responded to them all in the rebuttal.
> > > > > Specifically, for issue 1, **the environment is unknown** in our setting, which aligns with the extensive literature.
> > > > > For issue 2, it is not necessary to compare the learned reward model and the true reward model. The reason is that our objective is to learn a synthesis reward signal from the expert demonstration to induce the policy with better performance, **instead of recovering the true reward model.**
> > > > >
> > > > > And we iterate here again:
> > > > > * Our setting is the common setting in lots of IRL/MBIRL works (references in above reply), where only certain numbers of expert trajectories/behaviors are provided, and **rewards and dynamics/simulated environments are unknown**.
> > > > > * This paper focuses on recovering expert policy through the MCE IRL framework, which targets recovering expert policy under stochastic settings with provided expert trajectories[2,3]. Unlike world model learning task to recover reward model, which is essentially a supervised learning task[11,12], IRL inherently suffers from an ill-posed nature [1,2], meaning that for only a given set of expert behaviors, there can exist many different rewards that explain their optimality, making the problem fundamentally challenging. Thus, for evaluation, we only focus on whether our method can recover expert policy using provided expert trajectories, which is the same with literature work on IRL[2,3,4,13,14].
> > > > > * Our main contribution is proposing a model-based reward shaping to framework solve MCE IRL problem under stochastic setting. Consequently, **the theoretical bound presented in our paper concentrates on providing a theoretical framework to connect possible suboptimality of recovered policy with transition model learning error**. Thus, under this theoretical framework, there could be other possible algorithms with refinement or extension building upon our bound by using a more guaranteed accurate transition model or considering convergence and sampling errors from other policy optimization methods. We have empirically investigated the effect of different policy optimization methods under our model-enhanced reward shaping in the ablation study Appendix E.3.
> > > > >
> > > > > Thank you again for your feedback, and we'd love to address any further concerns.

---

> ### Comment · Area_Chair_9FEE · 2024-11-23
> **From AC.**
>
> Reviewer caVP : if possible, can you respond to the rebuttal?

---

> ### Author Response · Authors · 2024-11-27
>
> * Model-based RL is commonly employed in online settings, particularly with off-policy algorithms, as interactions with the real environment are crucial for ensuring the accuracy of the learned transition model. This accuracy, in turn, helps align the generated data samples with the true environmental dynamics [1,2]. While there are model-based approaches for offline RL, additional treatments are typically required to address the inaccuracies in transition learning and potential misalignment of generated samples [3]. Regardless of whether the setting is online or offline, model-based approaches generally do not assume prior knowledge about the transition dynamics [1], which is consistent with the assumptions made in our framework.
> * As highlighted in the examples above, different reward mechanisms can result in similar behavior or visitation distributions for trajectories. Consequently, we do not believe that any fair normalization technique exists to establish a relationship between fundamentally different reward mechanisms. The purpose of recovering a reward function in IRL is not to precisely replicate the true reward but to enable the agent to reproduce the same visitation distribution as the expert policy[4]. This alignment is typically demonstrated empirically through comparable levels of performance. If the reviewer considers the robot examples provided earlier, it becomes clear that entirely different reward mechanisms can lead to similar agent behaviors. Given this, there is no inherent relationship that must exist between such divergent reward functions. Moreover, pursuing such a comparison is beyond the scope of IRL, where the goal is not true reward learning, as would be the case in world model learning, but rather behavior replication.
>
>
> [1]Moerland, Thomas M., et al. "Model-based reinforcement learning: A survey." Foundations and Trends® in Machine Learning 16.1 (2023): 1-118.
>
> [2]Nair, Ashvin V., et al. "Visual reinforcement learning with imagined goals." Advances in neural information processing systems 31 (2018).
>
> [3]Yu, Tianhe, et al. "Mopo: Model-based offline policy optimization." Advances in Neural Information Processing Systems 33 (2020): 14129-14142.
>
> [4]Arora, Saurabh, and Prashant Doshi. "A survey of inverse reinforcement learning: Challenges, methods and progress." Artificial Intelligence 297 (2021): 103500.

---

### Official Review · Reviewer_Mu2o · 2024-10-31

**Soundness:** 2
**Presentation:** 3
**Contribution:** 2
**Rating:** 5
**Confidence:** 3

**Summary:**

This paper presents an enhanced version of Adversarial Inverse Reinforcement Learning (AIRL) by extending it to stochastic MDPs rather than deterministic ones. The authors revise the deterministic “next state” in AIRL to a probability distribution, which more accurately reflects real-world scenarios. The writing is clear and well-structured, making the paper easy to follow.

However, the key technique of transition modeling is not discussed, which could hinder reproducibility. Additionally, the experimental results are limited, making it challenging to fully assess the effectiveness of the proposed method.

**Strengths:**

1. The motivation for this paper is clearly presented. While most previous work focuses on deterministic MDPs, this framework addresses stochastic settings, which are more representative of real-world scenarios.
2. The paper is grounded in strong theoretical foundations, and all key arguments are thoroughly supported by theoretical verification.
3. The writing is clear and well-structured, making the paper easy to follow.

**Weaknesses:**

1. The main technical contribution of the proposed method is to incorporate system transition estimation into reward shaping for inverse reinforcement learning. However, the training process for the transition model is not explained in the paper. The method used to train this transition model and the accuracy of the model itself likely have a substantial impact on the final results. I suggest the authors provide additional analysis or details on this aspect.
2. It appears that the primary distinction between the proposed framework and AIRL is the incorporation of state transition probabilities within the reward function.
3. Experimental results are presented for only three environments, which limits the strength of the evaluation. Expanding the experiments to additional environments would make the results more convincing.

**Questions:**

Same as weakness.

Since the transition model training is not well-explained in the main paper, the results are hard to reproduce. I suggest the authors to provide source code.

---

> ### Author Response · Authors · 2024-11-18
> **Response to Reviewer Mu2o**
>
> We appreciate the comments provided by Reviewer Mu2o, and our responses are as follows.
>
> ### **Weakness 1:**
> The main technical contribution of the proposed method is to incorporate system transition estimation into reward shaping for inverse reinforcement learning. However, the training process for the transition model is not explained in the paper. The method used to train this transition model and the accuracy of the model itself likely have a substantial impact on the final results. I suggest the authors provide additional analysis or details on this aspect.
>
> **Answer:** We have incorporated implementation details in section 5.2 and Appendix F of the updated version. We also extend an ablation study on how the transition model learning error affects performance which can be found in Appendix E.2.
>
> ### **Weakness 2:**
> It appears that the primary distinction between the proposed framework and AIRL is the incorporation of state transition probabilities within the reward function.
>
> **Answer:** See General Response
>
> ### **Weakness 3:**
> Experimental results are presented for only three environments, which limits the strength of the evaluation. Expanding the experiments to additional environments would make the results more convincing. (Provide source code)
>
> **Answer:** We have conducted more experiments on more environments. And we also provide the source code to reproduce our results.

---

> ### Author Response · Authors · 2024-11-22
> **Reminder to Review Rebuttal**
>
> Dear Reviewer Mu2o,
>
> We understand this is a busy time of year, but we wanted to kindly remind you to review our rebuttal. Your feedback is invaluable for us to improve our paper. Please feel free to let us know if there are any specific points that require further clarification.
>
> Thank you for your time and effort. We truly appreciate your contributions to this process!
>
> Best,
> Authors

---

> > ### Comment · Reviewer_Mu2o · 2024-11-25
> > **Further Concerns**
> >
> > I acknowledge the authors for providing clarifications and conducting additional experiments. Despite these efforts, some concerns remain unaddressed:
> >
> > The manuscript appears to describe the incorporation of Gaussian noise into the observation space as a means to induce stochasticity. The authors then presuppose that the distribution of the transitioned states conforms to Gaussian distributions, which is a substantial assumption. Such an assumption may not be tenable in real-world scenarios. Typically, this type of problem is approached within the framework of Robust RL rather than stochastic processes. In Robust RL, various noise distributions are considered. What would be the approach if the distribution of the transition model were unknown? How would you model it?
> >
> > Moreover, the authors do not specify whether the introduced noise is applied to the raw observations or to normalized observations. In dimensions with large variances, the impact of added noise is markedly different from that in dimensions with smaller variances.
> >
> > Additionally, the training details for the transition models, particularly as outlined in Algorithm lines 6, and 10-11, have not been elucidated with sufficient clarity. Could you delineate the differences between the pretraining of the transition model and the updating of the dynamic model using Maximum Likelihood Estimation (MLE) loss? While I recognize that the authors have adopted most settings from Janner et al., 2019, it is important to note that their experiments were conducted in a deterministic environment.

---

> ### Comment · Area_Chair_9FEE · 2024-11-23
> **From AC.**
>
> Reviewer Mu2o: if possible, can you respond to the rebuttal?

---

> ### Author Response · Authors · 2024-11-25
> **Thanks for the getting back to us!**
>
> Thank you for your thoughtful response and questions! Below are our clarifications:
>
> 1. **Major Contribution and Transition Model Design**:
>    The assumption of Gaussian distributions has been adopted in several works within the field [7,8]. While we acknowledge that real-world applications often involve various types of noise, extending our transition model's robustness to handle different forms of uncertainty is an exciting direction for future work. Our primary contribution lies in introducing a **model-enhanced reward shaping framework to address the MCE IRL problem**. While designing a transition model that better captures stochasticity is indeed important, it is not the primary focus of this work. Nevertheless, many well-established model structures, such as RNN-based `RSSM` in Dreamer [1], `torchsde` [2,3], and VAE-like structures [5], could be seamlessly integrated into our framework to address more complex or high-dimensional dynamics. Given the challenges of learning a perfect transition model under diverse stochastic environments, our approach prioritizes practicality and computational efficiency. For this study, we opted for a simple transition model, which is sufficient to demonstrate the efficacy of our method while balancing computational resources and the extensive range of experiments conducted.
>
> 2. **Gaussian Noise and Normalization**:
>    Thank you for raising this point. In our current implementation, observations/environment objects are first normalized before Gaussian noise is applied. This ensures that the noise has a consistent effect across all dimensions, regardless of their original scale. Details of this implementation can be found in our anonymous code repository (linked in the General Response and Appendix F).
>
> 3. **Transition Model Training and Clarity**:
>    We appreciate the reviewer pointing out potential clarity issues regarding the transition model training. To clarify:
>    - Our approach differs from MBPO [4], which employs an ensemble of deterministic transition models with complex update rules. In contrast, we train a single transition model using a standard MLE loss for each update (as described in line 10 of our pseudocode).
>    - Line 11 explains the varying ratio of samples drawn from different buffers, which is detailed further in Appendix F.
>    - Regarding the pre-training phase, this is a standard procedure in off-policy algorithms to warm up the buffer [6]. Additionally, we update the transition model using the same MLE update loss in line 10 at each step during pre-training to mitigate potential divergence issues early in training. Detailed explanations can also be found in Appendix F.
>
> We thanks again for reviewer's thorough feedback, and we hope for your continued consideration of our work.
>
> [1]Hafner, Danijar, et al. "Dream to control: Learning behaviors by latent imagination." arXiv preprint arXiv:1912.01603 (2019).
> [2]Li, Xuechen, et al. "Scalable gradients for stochastic differential equations." International Conference on Artificial Intelligence and Statistics. PMLR, 2020.
> [3]Wang, Yixuan, et al. "Enforcing hard constraints with soft barriers: Safe reinforcement learning in unknown stochastic environments." International Conference on Machine Learning. PMLR, 2023.
> [4]Michael Janner, Justin Fu, Marvin Zhang, and Sergey Levine. When to trust your model: Model-based policy optimization. Advances in neural information processing systems, 32, 2019.
> [5]Girin, Laurent, et al. "Dynamical variational autoencoders: A comprehensive review." arXiv preprint arXiv:2008.12595 (2020).
> [6]Degris, Thomas, Martha White, and Richard S. Sutton. "Off-policy actor-critic." arXiv preprint arXiv:1205.4839 (2012).
> [7]Grunewalder, Steffen, et al. "Modelling transition dynamics in MDPs with RKHS embeddings." arXiv preprint arXiv:1206.4655 (2012).
> [8]Engel, Yaakov, Shie Mannor, and Ron Meir. "Reinforcement learning with Gaussian processes." Proceedings of the 22nd international conference on Machine learning. 2005.

---

> > ### Comment · Reviewer_Mu2o · 2024-11-26
> >
> > 1. I fully understand the main contribution of the work. However, utilizing a simple Gaussian distribution does not demonstrate the proposed scheme's effectiveness in more complex settings, especially when dealing with intricate distribution patterns. Your reference to a Gaussian setting dates back 12-20 years. It might be beneficial for the authors to explore more advanced tools, such as diffusion models, and consider uncertainties arising from system dynamics, beyond merely adding Gaussian random noise.
> >
> > 2. I did not find any normalization operations in your source code. Whether you have normalized the observations and actions or not, it is important to emphasize this in your paper to prevent potential misunderstandings.
> >
> > Given the feedbacks from other reviewers, I will retain my score and level of confidence.

---

### Official Review · Reviewer_qE6H · 2024-11-03

**Soundness:** 3
**Presentation:** 2
**Contribution:** 2
**Rating:** 5
**Confidence:** 4

**Summary:**

The paper introduces a model-enhanced adversarial inverse reinforcement learning (IRL) approach that integrates transition model estimation into reward shaping, with theoretical guarantees and superior performance in stochastic environments.

**Strengths:**

1. The paper presents a incremental framework over adversirial inverse RL that integrates model-based transition dynamics with adversarial IRL, potentially addressing performance degradation in stochastic environments, which is a good direction because considering the stochasticity in the environments is significant.
2. The authors show some theoretical guarantees of the optimality of the method and the transition model error.

**Weaknesses:**

1. The writing quality of the paper requires improvement. For instance, while the core idea is relatively straightforward, the title is overly long and complex, making it challenging to grasp before reading. The authors should also enhance the clarity of propositions and theorems by providing more detailed descriptions and insights, explicitly connecting them to the paper’s main ideas. Additionally, the numbering of theorems and their proofs in the appendix should be consistent and aligned for easier reference.
2. The paper modifies the reward shaping term from $\phi(s_t)$ to $E_{\mathcal{T}}[\phi(s_{t+1})|s_t, a_t]$, incorporating the transition model into the reward shaping to better handle stochastic environments. While the idea is conceptually straightforward, it introduces additional complexity in modeling the transition function, especially in real-world environments with inherent stochasticity in transitions.
3. In Section 5.3, the authors present a theoretical analysis of performance. However, the analysis appears to apply broadly to inverse RL settings with transition mismatches, offering limited theoretical insight into the specifics of the proposed method. In particular, Theorems 5.3 and 5.4 provide bounds on rewards and value functions across two MDPs, yet they do not address the optimality or performance bounds of the policy learned through the proposed approach. This generality detracts from a deeper understanding of how the method uniquely impacts policy performance.
4. The authors claim that their method achieves improved sample efficiency, as demonstrated in Section 6 (line 439). However, it is unclear if the authors have accounted for the additional computational costs involved in modeling the transition functions. Moreover, it remains ambiguous whether the observed sample efficiency in policy learning is primarily due to the proposed method itself or the specific techniques introduced in Section 5.2 (line 316). Clarifying these aspects would strengthen the evaluation of the method’s true efficiency gains.
5. The experiments are too limited to thoroughly evaluate the effectiveness of the proposed method. Specifically,
- The authors conduct experiments only on three relatively simple, low-dimensional environments and compare against just three baseline methods (see Tables 2 and 3). To strengthen the evaluation, the authors should consider testing on more challenging environments and expanding the comparisons to include additional algorithms, such as SAM, SQIL, and GAIfO, as mentioned in Table 1.
- Appendix D lists a few key hyperparameters, but several essential details are missing, which could significantly hinder reproducibility. Specifically, omitted hyperparameters include the batch size, discount factor, the number of discriminator training epochs per policy update, and exact 5 random seed values, etc. Additionally, while sharing code is not mandatory, its absence makes it even more challenging to accurately replicate the experimental setup and results.
- The authors should conduct more ablation studies. For example, with different noise scale in stochastic environments, to show the robustness of the proposed method.
- In Figure 4, both DAC and the proposed method show non-zero initial performance, which raises questions about the setup or initialization of these models. Additionally, it’s unclear why both methods terminate early in Figure 4, whereas they continue for the full duration in Figures 2 and 3.

**Questions:**

Please see the above.

---

> ### Author Response · Authors · 2024-11-18
> **Response to Reviewer qE6H [1/2]**
>
> We appreciate reviewer qE6H's comments. Following are our response.
>
> ### **Weakness 1:**
> The writing quality of the paper requires improvement. For instance, while the core idea is relatively straightforward, the title is overly long and complex, making it challenging to grasp before reading. The authors should also enhance the clarity of propositions and theorems by providing more detailed descriptions and insights, explicitly connecting them to the paper’s main ideas. Additionally, the numbering of theorems and their proofs in the appendix should be consistent and aligned for easier reference.
>
> **Answer:** We have made changes accordingly including title in the updated paper. Changes are highlighted in cyan color.
>
>
> ### **Weakness 2:**
> The paper modifies the reward shaping term from $\phi(s_t)$ to $E_\tau [\phi(s_{t+1})\mid s_t,a_t]$, incorporating the transition model into the reward shaping to better handle stochastic environments. While the idea is conceptually straightforward, it introduces additional complexity in modeling the transition function, especially in real-world environments with inherent stochasticity in transitions.
>
> **Answer:** In our setting, we only need to train a simple transition model as a two-layered low-dimensional MLP which is already sufficient to predict a few step horizon. Thus, we believe that comparing to the performance and sample efficiency improvement brought by this simple structure, which has been thoroughly investigated in our experiments, a little additional complexity is cost-effective.
>
>
> ### **Weakness 3:**
> In Section 5.3, the authors present a theoretical analysis of performance. However, the analysis appears to apply broadly to inverse RL settings with transition mismatches, offering limited theoretical insight into the specifics of the proposed method. In particular, Theorems 5.3 and 5.4 provide bounds on rewards and value functions across two MDPs, yet they do not address the optimality or performance bounds of the policy learned through the proposed approach. This generality detracts from a deeper understanding of how the method uniquely impacts policy performance.
>
> **Answer:** Our main contribution is proposing a model-based reward shaping to framework solve MCE IRL problem under stochastic setting. Consequently, the theoretical bound presented in our paper concentrates on providing a theoretical framework to connect possible suboptimality of recovered policy with transition model learning error. Thus, under this theoretical framework, there could be other possible algorithms with refinement or extension building upon our bound by using more guaranteed accurate transition model or considering convergence and sampling errors from other policy optimization methods.
>
>
> ### **Weakness 4:**
> The authors claim that their method achieves improved sample efficiency, as demonstrated in Section 6 (line 439). However, it is unclear if the authors have accounted for the additional computational costs involved in modeling the transition functions. Moreover, it remains ambiguous whether the observed sample efficiency in policy learning is primarily due to the proposed method itself or the specific techniques introduced in Section 5.2 (line 316). Clarifying these aspects would strengthen the evaluation of the method’s true efficiency gains.
>
> **Answer:** We have added an ablation study in Appendix E.3 investigating the performance difference between DAC, model-enhanced reward shaping with SAC, and our method. Our result shows that our model-enhance reward shaping technique significantly contributes to the sample efficiency and performance improvement. Model-based policy optimization techniques further improve the sample efficiency, however, it doesn't cause notable differences in terms of performance, which makes sense since both policy optimization methods should converge to the same optimality[1].

---

> > ### Author Response · Authors · 2024-11-18
> > **Response to Reviewer qE6H [2/2]**
> >
> > ### **Weakness 5:**
> > The experiments are too limited to thoroughly evaluate the effectiveness of the proposed method. Specifically:
> > * The authors conduct experiments only on three relatively simple, low-dimensional environments and compare against just three baseline methods (see Tables 2 and 3). To strengthen the evaluation, the authors should consider testing on more challenging environments and expanding the comparisons to include additional algorithms, such as SAM, SQIL, and GAIfO, as mentioned in Table 1.
> > * Appendix D lists a few key hyperparameters, but several essential details are missing, which could significantly hinder reproducibility. Specifically, omitted hyperparameters include the batch size, discount factor, the number of discriminator training epochs per policy update, and exact 5 random seed values, etc. Additionally, while sharing code is not mandatory, its absence makes it even more challenging to accurately replicate the experimental setup and results.
> > * The authors should conduct more ablation studies. For example, with different noise scale in stochastic environments, to show the robustness of the proposed method.
> > * In Figure 4, both DAC and the proposed method show non-zero initial performance, which raises questions about the setup or initialization of these models. Additionally, it’s unclear why both methods terminate early in Figure 4, whereas they continue for the full duration in Figures 2 and 3.
> >
> > **Answer:**
> > - We have updated the experiment section with extra environments and extensive investigations. Please refer to general response and updated paper for detailed description.
> >     - SQIL[2] is built upon Q-Learning and is specifically designed for discrete action space, making it unsuitable for current continuous setting.
> >     - GAIfO is an extension of GAIL and has similar performance as GAIL.  It is primarily designed to handle high-dimensional image observation space[6], thus direct comparison with GAIL is sufficient.
> >     - Existing comparisons between SAM and DAC in MuJoCo[4,5] indicate that DAC outperforms SAM in both performance and sample efficiency. Hence, a comparison with SAM is unnecessary. Additionally, all the aforementioned methods are evaluated under deterministic environments, which differ from our focus on stochastic settings.
> > - We have extended the implementation details in the appendix and provided the source code to reproduce our results in Appendix F of the updated paper.
> > - We have conducted additional ablation studies to further analyze our method. Details are included in the general response and Appendix E of the updated paper.
> > - Figures 2, 3, and 4 from the original paper have been replaced with more comprehensive and structured training graphs, now located in Appendix D and updated Figure 2. All policies use the same structure and initialization procedures across the compared methods. Note that the non-zero performance observed in the graphs reflects the first evaluation step, as the policies undergo initial training for a set number of steps before evaluation. The training graphs report the average performance at each evaluation step.
> >
> >
> > [1]Michael Janner, Justin Fu, Marvin Zhang, and Sergey Levine. When to trust your model: Model-based policy optimization. Advances in neural information processing systems, 32, 2019.
> >
> > [2]Reddy, Siddharth, Anca D. Dragan, and Sergey Levine. "Sqil: Imitation learning via reinforcement learning with sparse rewards." arXiv preprint arXiv:1905.11108 (2019).
> >
> > [3]Blondé, Lionel, and Alexandros Kalousis. "Sample-efficient imitation learning via generative adversarial nets." The 22nd International Conference on Artificial Intelligence and Statistics. PMLR, 2019.
> >
> > [4]Orsini, Manu, et al. "What matters for adversarial imitation learning?." Advances in Neural Information Processing Systems 34 (2021): 14656-14668.
> >
> > [5]Arulkumaran, Kai, and Dan Ogawa Lillrank. "A pragmatic look at deep imitation learning." Asian Conference on Machine Learning. PMLR, 2024.
> >
> > [6]Faraz Torabi, Garrett Warnell, and Peter Stone. Generative adversarial imitation from observation. arXiv preprint arXiv:1807.06158, 2018b.

---

> > > ### Comment · Reviewer_qE6H · 2024-11-23
> > > **Response to authors**
> > >
> > > Thanks for the authors' reply.
> > >
> > > The response addressed some of my concerns, and I raised my score from 3 to 5 accordingly. However, I believe the paper is undergoing significant revisions compared to the initial one, especially the extensive additional experiments. After considering the comments from other reviewers, my overall stance remains negative.

---

> ### Author Response · Authors · 2024-11-22
> **Reminder to Review Rebuttal**
>
> Dear Reviewer qE6H,
>
> We understand this is a busy time of year, but we wanted to kindly remind you to review our rebuttal. Your feedback is invaluable for us to improve our paper. Please feel free to let us know if there are any specific points that require further clarification.
>
> Thank you for your time and effort. We truly appreciate your contributions to this process!
>
> Best,
> Authors

---

> ### Author Response · Authors · 2024-11-23
> **Thanks for getting back to us!**
>
> Thank you for taking the time to review our rebuttal and for acknowledging the improvements we made to address your concerns. We sincerely appreciate your effort in carefully considering our revisions and providing valuable feedback.
>
> We would like to clarify that the additional experiments were conducted primarily to further validate the **performance superiority, robustness, and sample efficiency of our novel off-policy model-enhanced reward shaping framework**, which addresses the Maximum Causal Entropy IRL problem in **stochastic settings**. Importantly, the **core methodology, theoretical contributions, and key statements regarding efficacy and efficiency** remain unchanged from the initial submission. The additional experiments were specifically included to address the feedback received during the review process and provide stronger empirical support for our claims, not to alter the foundational aspects of the paper.
>
> We believe we have thoroughly addressed all the concerns raised during the rebuttal period. However, we respect your overall stance and deeply value your perspective. If there are any specific aspects of the paper that remain unclear or insufficient, we would be happy to provide further clarifications or address those points directly.
>
> Thank you again for your thoughtful feedback, and we hope for your continued consideration of our work.

---

### Official Review · Reviewer_PcDm · 2024-11-04

**Soundness:** 2
**Presentation:** 3
**Contribution:** 2
**Rating:** 6
**Confidence:** 2

**Summary:**

The paper proposes a model-enhanced reward-shaping method for adversarial inverse reinforcement learning in the stochastic environment where AIRL is likely to fail. By integrating the transition model to the account for the stochastic transition. Also, they provide an error bound for it.

**Strengths:**

1, Propose a method that tackles the limitations of existing methods in the stochastic environment.

2, The method is simple and intuitive.

3, The error bound in the paper is well connected with the paper.

**Weaknesses:**

1, Experiments are only in Hopper and inverted pendulum, while the latter seems to be an easy setting for all the methods.

2, Some of the presentation is not clear. For example, the L11 in the algorithm is confused, Does it correspond to the L324-331? And why do you sample H-step trajectories (can you connect it with the sample efficiency somewhere )?

**Questions:**

1, Can you explain intuitively why not having the expectation in Eq(6) doesn't work?

2. Why compare with a deterministic environment? Isn't it the same with other reward shaping method?

3, Why the L11 in algorithm will mitigate the distribution shift? Can you give a little bit more background on why this induces distribution shift?

4, I am wondering can this idea be applied to the GAIL, which uses the transition model to generate a mixture of the D_{gen} and D_{env}? And also account for the stochastic of the env in the -logD for gail?

5, How do you vary the proportion of selecting synthetic data or environment data for policy optimization.  And how do you select H? And together how they affect the sample efficiency?

6, If the transition model is unbounded, for example, when the transition model is hard to learn in more complex environments, can it still outperform AIRL? I believe more samples will be needed.

---

> ### Author Response · Authors · 2024-11-18
> **Response to Reviewer PcDm [1/2]**
>
> We thank Reviewer PcDm for the comment. Following are our answers to the mentioned questions and concerns.
>
> ### **Weakness 1:**
> Experiments are only in Hopper and inverted pendulum, while the latter seems to be an easy setting for all the methods.
>
> **Answer:** We have extended our experiments to additional environments (e.g., HalfCheetah-v4 and Walker2d-v4) with additional exploration on the effects of quantity of expert trajectories and a set of ablation studies. Please check the detailed illustration in General Response and our updated paper.
>
> ### **Weakness 2:**
> Some of the presentation is not clear. For example, the L11 in the algorithm is confused, Does it correspond to the L324-331? And why do you sample H-step trajectories (can you connect it with the sample efficiency somewhere)?
>
> **Answer:** We have fixed a few presentation issues in the updated paper. Line 11 in the algorithm does correspond with Sample Efficiency section. Generating $H$ steps of trajectories from the learned transition model is a common approach for Model-based RL [1,2,3,4] to supplement real environmental samples for policy optimization.
>
> ### **Question 1:**
> Can you explain intuitively why not having the expectation in Eq(6) doesn't work?
>
> **Answer:** The reasons here are two-fold. First, let's consider a stochastic MDP with continuous state $\mathcal{S}$ and action $\mathcal{A}$ space, which means $\mathcal{T}(s_{t+1}|s_t,a_t)\rightarrow[0,1]$, where $s_t,s_{t+1}\in\mathcal{S}, a_t\in\mathcal{A}$, and $\vert\mathcal{S}\vert\rightarrow\infty,  \vert\mathcal{A}\vert\rightarrow\infty$. The intuitive explanation is that each $(s_t,a_t,s_{t+1})$ tuple can be regarded as a sample in a continuous distribution. Since state space and action space are continuous, there are numerous possible tuples requiring extensive sampling to recover the reward. Secondly, for different tuples above, each has a transition probability. There could be a scenario that we recover high reward to a tuple with extremely low transition probability, which in reality might be very unlikely to reach. This is called risk-seeking behavior[5], commonly caused by directly shifting ME-based algorithms to stochastic environments.
>
> ### **Question 2:**
> Why compare with a deterministic environment? Isn't it the same with other reward shaping method?
>
> **Answer:** The reason for comparing under deterministic settings is to validate that our method has competitive performance with all the other baselines. And in the stochastic settings, our method can achieve much better performance compared to baselines.
>
> ### **Question 3:**
> Why the L11 in algorithm will mitigate the distribution shift? Can you give a little bit more background on why this induces distribution shift?
>
> **Answer:** To clarify that Line 11 in the algorithm does not mitigate the distribution shift, we introduce the distribution shift mitigation method in lines 327 to 333 in the updated paper version. The reason for the distribution shift is that at the early stage of training, distribution of learned transition model $\hat{\mathcal{T}}$ is far from real $\mathcal{T}$, thus policy optimization based on samples from both transition dynamic might cause deviation without any treatment[1], since part of the samples might be unrealistic under real environment.
>
> ### **Question 4:**
> I am wondering can this idea be applied to the GAIL, which uses the transition model to generate a mixture of the D_{gen} and D_{env}? And also account for the stochastic of the env in the -logD for gail?
>
> **Answer:** For GAIL, it is a direct imitation learning algorithm, thus it is not recovering reasonable rewards as IRL. Therefore, the learned transition model might not be helpful to GAIL. Additionally, GAIL is an on-policy algorithm, and model-based policy optimization approaches normally are off-policy, thus learned transition model might not be helpful on both end.
>
> ### **Question 5:**
> How do you vary the proportion of selecting synthetic data or environment data for policy optimization. And how do you select H? And together how they affect the sample efficiency?
>
> **Answer:** At the beginning of the training, the proportion to select synthetic data is 0, since the learned transition model is not accurate at this point. As training progresses, the proportion increases linearly with the training environmental steps. For H, we set it to 2, but from past experience anything below 8 should work. Details of these hyper-parameters selection schemes can be found in Appendix F. Generally, increased proportion and H will result in more trajectories generated. However, depending on the learned transition model capacity, as H increases, the generated trajectories might become less realistic, resulting in suboptimal policy performance.

---

> > ### Author Response · Authors · 2024-11-18
> > **Response to Reviewer PcDm [2/2]**
> >
> > ### **Question 6:**
> > If the transition model is unbounded, for example, when the transition model is hard to learn in more complex environments, can it still outperform AIRL? I believe more samples will be needed.
> >
> > **Answer:** In this case, more powerful parameterized functions are needed to model the transition dynamic, since the learned transition model plays a significant role in both rewards shaping and policy optimization. We extend an ablation study to provide empirical evaluation on how transition model learning error affects performance in Appendix E.2.
> >
> >
> > [1]Michael Janner, et al. When to trust your model: Model-based policy optimization. Advances in neural information processing systems, 32, 2019.
> >
> > [2]Sinong Zhan, et al. State-wise safe reinforcement learning with pixel observations. In 6th Annual Learning for Dynamics and Control Conference, pp. 1187–1201. PMLR, 2024.
> >
> > [3]Nicklas Hansen, et al. Temporal difference learning for model predictive control. arXiv preprint arXiv:2203.04955, 2022.
> >
> > [4]Danijar Hafner, et al. Dream to control: Learning behaviors by latent imagination. arXiv preprint arXiv:1912.01603, 2019.
> >
> > [5]Brian D Ziebart, et al. Modeling interaction via the principle of maximum causal entropy. 2010.

---

> ### Author Response · Authors · 2024-11-22
> **Reminder to Review Rebuttal**
>
> Dear Reviewer PcDm,
>
> We understand this is a busy time of year, but we wanted to kindly remind you to review our rebuttal. Your feedback is invaluable for us to improve our paper. Please feel free to let us know if there are any specific points that require further clarification.
>
> Thank you for your time and effort. We truly appreciate your contributions to this process!
>
> Best,
> Authors

---

> ### Comment · Area_Chair_9FEE · 2024-11-23
> **From AC.**
>
> Reviewer PcDm: if possible, can you comment on the rebuttal?

---

### Author Response · Authors · 2024-11-18
**General Response**

We thank all the reviewers for the effort they put in to improve our paper from various perspectives. This work aims to propose a novel model-based reward-shaping method to solve Maximum Causal Entropy IRL under stochastic settings. Below is our general response. First, we want to clarify that adversarial IRL/IL[1] is a category including all methods formulating the imitation learning problem into an adversarial training structure, and **AIRL[2] is a specific deterministic on-policy algorithm based on Maximum Entropy (ME) framework** falling into that category, so do plenty of others. To clarify, our approach is an **off-policy algorithm targeting stochastic setting** with an adversarial training framework as all the other adversarial IL/IRL works, but it is rooted in the **Maximum Causal Entropy (MCE) framework**, rather than the standard ME followed by AIRL, DAC, etc, leading to distinctive discriminator formulation and training scheme. The ME framework maximizes the entropy of a policy to encourage exploration without regard to environmental stochasticity, while the MCE framework specifically maximizes entropy over action sequences conditioned on past states and actions, ensuring that the policy accounts for the stochastic nature of the environment. We illustrate the difference between MCE and ME frameworks and resulting mathematical IRL problem formulations in lines 171-183. More details can be found in [3,4]. This nature together with **model-based off-policy** training fashion leads to our method's efficacy in performance and sample efficiency under stochastic settings, which we further validate this statement in our newly added experiments introduced in the next paragraph. We provide the theoretical insight in Proposition 5.1 to show the equivalence between our adversarial training objective and MCE objective in reward learning.

**_We acknowledge that the original experiments and algorithmic details are not sufficient, thus we have updated our paper with more thorough experiments under stochastic settings, ablation studies, and algorithmic details including hyperparameters selection and source code (See newly updated PDF)._**  Detailed changes are in the following sections:
* [**Implementation Details and Source Code**] We have updated the algorithmic details of our method, including transition model learning and implementation specifics, which are now available in Section 5.2 (lines 306-310) and Appendix F (page 26). Additionally, we have included an anonymous link to our code repository in Appendix F, which contains the expert policies used for expert trajectory collections, as well as implementations of baseline methods and our proposed approach. (https://anonymous.4open.science/r/MBIRL-4C2F/README.md)
* [**More Environments**] We have added new experiments in the HalfCheetah-v4 and Walker2d-v4 environments under stochastic and deterministic conditions. Our experiments also support our claims of competitive performance under deterministic settings and superior performance under stochastic settings. These results are provided in lines 428-469 and lines 501-511.
* [**Sample Efficiency**] To examine the sample utilization efficiency of provided expert trajectories under stochastic settings, we conducted tests across all five environments using varying numbers of expert trajectories, from 5 to 1000. Our method demonstrated notable performance superiority across most environments, regardless of the quantity of expert trajectories provided. Results can be found in lines 471-485 and provided training curves of all environments can be found in Appendix D. The results indicate that our method consistently recovers expert-level performance even with limited expert trajectories, a capability that all baselines miss in such settings. Furthermore, as shown in Appendix D, our method also achieves expert-level performance with the fewest environmental steps in all environments.
* [**Ablation Studies**] We conducted three additional ablation studies, presented in Appendix E. The first study examines our method's robustness to different levels of stochasticity. The second study explores the effects of transition model learning errors on reward learning. The third study investigates under the same model-enhanced reward shaping formulation, how SAC and Model-based Policy Optimization methods affect the performance and sample efficiency differently.


[1]Orsini, Manu, et al. "What matters for adversarial imitation learning?." Advances in Neural Information Processing Systems 34 (2021): 14656-14668.

[2]Fu, Justin, et al. "Learning robust rewards with adversarial inverse reinforcement learning." arXiv preprint arXiv:1710.11248 (2017).

[3]Brian D Ziebart, et al. Modeling interaction via the principle of maximum causal entropy. 2010.

[4]Gleave, Adam, et al. "A primer on maximum causal entropy inverse reinforcement learning." arXiv preprint arXiv:2203.11409 (2022).

---

### Meta-Review · Area_Chair_9FEE · 2024-12-20

**Metareview:**

This paper addresses adversarial inverse reinforcement learning (IRL), with a view of supporting stochastic environments. To accomplish this, it uses reward shaping based on a learned transition model.

The main strengths of the paper are that (1) stochastic environments are important in practice and (2) the approach has thoretical justification.

However, important concerns remain. Specifically:
- the experiments are still limited, even after having been expanded in the rebuttal phase
- using performance profiles (https://agarwl.github.io/rliable/) would be nice for this type of paper
- the writing and clarity needs to improve by a lot

For these reasons, I recommend rejection.

**Additional Comments On Reviewer Discussion:**

All reviewers agree that the experiments are insufficient. The only review recommending acceptance is the lowest-quality one.

---

### Decision · Program_Chairs · 2025-01-22

Reject